# Partitioning Entropy with Action Mechanics: Predicting Chemical Reaction Rates and Gaseous Equilibria of Reactions of Hydrogen from Molecular Properties

**DOI:** 10.3390/e23081056

**Published:** 2021-08-16

**Authors:** Ivan R. Kennedy, Migdat Hodzic

**Affiliations:** 1School of Life and Environmental Sciences, Sydney Institute of Agriculture, University of Sydney, Sydney, NSW 2006, Australia; 2Faculty of Information Technologies, The Dzemal Bijedic University of Mostar, 88000 Mostar, Bosnia and Herzegovina; migdathodzic@gmail.com

**Keywords:** entropy, enthalpy, equilibrium, ergal, free energy, Gibbs energy, Haber process, Helmholtz energy, hydrogen dissociation, transition state, water dissociation, virial theorem

## Abstract

Our intention is to provide easy methods for estimating entropy and chemical potentials for gas phase reactions. Clausius’ virial theorem set a basis for relating kinetic energy in a body of independent material particles to its potential energy, pointing to their complementary role with respect to the second law of maximum entropy. Based on this partitioning of thermal energy as sensible heat and also as a latent heat or field potential energy, in action mechanics we express the entropy of ideal gases as a capacity factor for enthalpy plus the configurational work to sustain the relative translational, rotational, and vibrational action. This yields algorithms for estimating chemical reaction rates and positions of equilibrium. All properties of state including entropy, work potential as Helmholtz and Gibbs energies, and activated transition state reaction rates can be estimated, using easily accessible molecular properties, such as atomic weights, bond lengths, moments of inertia, and vibrational frequencies. We conclude that the large molecular size of many enzymes may catalyze reaction rates because of their large radial inertia as colloidal particles, maximising action states by impulsive collisions. Understanding how Clausius’ virial theorem justifies partitioning between thermal and statistical properties of entropy, yielding a more complete view of the second law’s evolutionary nature and the principle of maximum entropy. The ease of performing these operations is illustrated with three important chemical gas phase reactions: the reversible dissociation of hydrogen molecules, lysis of water to hydrogen and oxygen, and the reversible formation of ammonia from nitrogen and hydrogen. Employing the ergal also introduced by Clausius to define the reversible internal work overcoming molecular interactions plus the configurational work of change in Gibbs energy, often neglected; this may provide a practical guide for managing industrial processes and risk in climate change at the global scale. The concepts developed should also have value as novel methods for the instruction of senior students.

## 1. Introduction

This investigation aims to clearly identify neglected variables in chemical process theory and then to promote their application to catalysis and the achievement of states of minimum Gibbs energy. The main variable is the thermal energy stored as non-sensible heat of elevated quantum states in molecular systems, particularly by gases. This is relevant to explaining the principle of maximum entropy characteristic of most Earth systems and also of least variation in action, as described by Lagrangian theory. Our study also seeks to show how to calculate absolute values for the action potentials governing these states. In particular, we examine how action mechanics [1,2,3,4] may help provide better understanding of catalysis, speeding up chemical processes’ achievement of equilibrium and maximum entropy.

The concept of action space [d*mr**ω*.d*r**φ*, Joule.sec] was proposed [1,2,3,4] in action mechanics as a more accessible surrogate for the phase-space [d*p*.d*r*, J.sec] of statistical mechanics. Its significance is that all matter exists in action space, including the primary particles or molecules, which are sustained in real-time Brownian motion by impetus from resonant radiant energy [Σ*mc*^2^, J] in the field [1]. Following Einstein, who proposed [5] such a mobilising force for Brownian motion, we can also assume that matter gains potential energy [*V*, J] from this field, which is formally equivalent to elevated quantum states. Given that kinetic energy *T* is ½*mv*^2^ for particles of velocity *v*, then we have [1] total energy equal to *mc*^2^ − ½*mv*^2^ [E, J], which is a conserved property as the first law of thermodynamics.
*V* = *mc*^2^ − *mv*^2^(1)

Then, the total energy *E* can be given.
*E* = *mc*^2^ − ½*mv*^2^(2)

### 1.1. The Virial Theorem

Classically, prior to Einstein’s theories of special and general relativity, the base level for potential energy (Equation (1)) omitted the *mc*^2^ term involving the invariant speed of light *c*, setting the base potential energy (*V*) as zero by convention. A need for clarity regarding energy and the first law of thermodynamics prompted Clausius [6] to define the virial theorem relating the mean kinetic energy [*T* = Σ*mv*^2^/2, J] of a system of particles to the mean potential energy [*V* = Σ(−*mv*^2^), J]. This theorem should not be confused with the virial expansion that refers to a mathematical expression for the departure of gases from ideality, indicating their degree of molecular interaction. When considering particles as material points at common gravitational potential, Clausius defined an internal work function (*W*) he termed the ergal [J] as the following expression in rectangular coordinates [*x*, *y*, *z*], where *X*, *Y*, and *Z* were Cartesian components of a central force field exerted on the particles [*X = md*^2^*x*/*dt*^2^, *Y = md*^2^*y*/*dt*^2^, *Z = md*^2^*z*/*dt*^2^, Newtons]; he had sought to find a short alternative term to Rankine’s new definition of potential energy.
(3)$$W=\Sigma \left(Xdx+Ydy+Zdz\right)$$

He distinguished internal work within the body from external work as subject to different conditions and also equated the variation in the ergal or internal work to variation in the internal kinetic energy.
(4)$$d\Sigma m\nu^{2}/2=\Sigma \left(Xdx/dt+Ydy/dt+Zdz/dt\right)dt$$

We remark that this thermal variation in the vis viva or kinetic energy was thus equated to the variation of the power expressed by the ergal, which was integrated with time as separate and complementary forms of energy. 

“The work done during any time by the forces acting upon a system is equal to the increase in the vis-viva of the system during the same time”. Clausius [7] separated the ergal from kinetic energy as a function of the integrated force acting in the system of material particles, indicated by the letter *J*. In the Mathematical Introduction of his text on The Mechanical Theory of Heat, he related the force originating in both attractions and repulsions exerted on the moving points as depending on distance and therefore acting as central force.
(5)$$\Sigma \left(Xdx+Ydy+Zdz\right)=-dJ$$

Thus, the kinetic work done in any time could be regarded as equal to the decrease in the potential given by the ergal. He further defined that “The sum of the vis-viva and the ergal remains constant during the motion”.
(6)U=T+J

Here, *U* [J] was regarded as the total internal energy of the system, understood as the principle of the conservation of energy.

### 1.2. Two Kinds of Virial Effects 

We can distinguish two separate applications of the virial theorem since Clausius’ definition.
(i)The virial theorem has been adopted by physicists for systems of particles such as the hydrogen atom responsible for the solar spectrum and the gravitational structure of stars. The theorem defines the kinetic and potential energy of particles, for variations involving absorption or emission of quanta travelling at light speed. For verification of modelling in quantum theory, variations in kinetic energy must be set equal to the energy of the quanta, with their absorption resulting in increased potential energy equal to the decrease in kinetic energy. So, the change in potential energy is considered as equal to the sum of the decrease in the kinetic energy plus the quantum absorbed into the local field. Considering a single particle subject to quantum excitation [Σ*hv*, J] in a central field, the variation in potential energy can be expressed as the sum of the quanta absorbed plus the decrease in the kinetic energy.
(7)Σhν+−δT=δVThe variation in potential energy of collapsing stars can be expressed by the same equation in reverse, leading to emission of radiation with an equivalent increase in kinetic energy and temperature of the stars. We have claimed elsewhere [8] that molecules in the Earth’s atmosphere are subject to this form of the virial theorem, yielding a lapse rate for change of temperature with altitude from balancing thermal and gravitational energies of 6.9 K per km from the land surface, which is consistent with observations, a gradient far from isothermal, nor the result of adiabatic expansion.(ii)By contrast, in systems of molecules moving as material points in a heat bath providing a multiplicity of central forces, as in an ideal gas, the relationship between kinetic and potential energy differs as a function of temperature and pressure. This is illustrated in the case of water where the molecules can exist frozen at low temperature, aggregated as clusters in liquid able to flow under gravity, and as separate molecules in the gas or vapor phase. All three forms exist on the Earth’s surface, in contrast with all other gaseous constituents of the atmosphere, only existing as vapor. Heat transmitted as quanta does reversible work on all three phases of water, melting or vaporising water while raising the temperature of the gas. For systems in a common heatbath, the absorption of quanta no longer results in decreased kinetic energy and temperature of orbiting particles, in contrast to matter bound by a dominant central force, such as electrons in atoms or matter by gravity. Clausius’ Equation (8) describes [6] the reversible effect of heat on the system. This equates the effect of heat d*Q* [J] on the variation in the quantity of sensible heat in the body d*H* plus the internal work or potential of the ergal d*J* and the external work d*W*.
d*Q* = d*H* + d*J* + d*W*(8)

For liquid water, d*H* denotes changes in the sensible heat content of molecules as shown by their temperature; d*J* refers to heat needed to disaggregate water molecules from H-bonded clusters and for expansion some 1630 times while doing work d*W* against atmospheric pressure. Alternatively, the work of expansion is against the much greater back pressure reversibly exerted by a heat engine as in the Carnot cycle [4]. To simplify the equation on the grounds that only the sum of *H* and *J* could be measured, Clausius combined them as *U*, referring to this as the internal Energy of the body of particles. Here, d*Q* clearly refers to the sum of changes in both kinetic and potential forms of energy.
d*Q* = d*U* + d*W*(9)

However, we will show that there is a possible flaw in combining the ergal in *U* as energy specific to a particular system of molecules. Their configuration may also include different potentials for internal work that is required to sustain external work, which is consistent with Newton’s law that for every action, there is an equal and opposite reaction. We suggest that this flaw is the neglect of the internal work of increasing quantum states of molecules for vibration, rotation, and translation.

### 1.3. Some Lessons from Revising the Carnot Cycle

In an action revision of the Carnot cycle [4], we have shown how the negative Gibbs energy of the working fluid could be equated with Carnot’s caloric. As quanta, that field energy can sustain the configuration of the working fluid as material particles while expanding isothermally, supports Carnot’s opinion that the specific heat is also logarithmically variable with the change of volume of the working fluid (Figure 1). By including this internal work supporting external work in the term specific heat, Carnot perhaps extended this terminology too far. Furthermore, the heat required by the theorem is solely a function of the increased ratio of the volumes of gas rather than their mass. The extra heat required in the isothermal system means that the total internal energy of the system can be varied. Then, the maximum work possible from a fully reversible cycle is equal in Carnot’s terms to the difference between the caloric absorbed isothermally by the working fluid at a high temperature in Stage 1 of the cycle and the lesser caloric absorbed isothermally by the low temperature sink in Stage 3. These same amounts of heat are logarithmic functions of changes in volume of the working fluid at the higher temperature and to the change in volume at the lower temperature. This difference could equally well be obtained from the inverse ratio of internal pressures in the cylinder at the high and low temperatures. We showed how these reversible processes affected the Gibbs energy of the working fluid, and so they correspond to the second part of the ergal as defined by Clausius with the isothermal processes. When steam was generated from water, the heat of dissociation of the water clusters into separate molecules would comprise the non-ideal part of the gas behaviour, though for the permanent gases argon and nitrogen in the Carnot cycle, this part of the ergal as significant aggregation is absent. 

In this paper, we reinterpret the role of the ergal in modern terms as including this requirement for thermal energy as virtual quanta in the field supporting the internal work of sustaining the configuration of the molecules. That there is equivalence between changes in kinetic energy and the configurational ergal does not mean that added heat is consumed as the kinetic energy; it is partly required as a form of energy sustaining the configuration of the system, supporting that kinetic heat of molecular motion as a complementary function. This is important as the pressure exerted by the material points in the piston is required by the system to perform external work. Cylinders in Figure 1 containing only the quantum energy of the ergal could do no work, given the vast diminution in the rate of action impulses in a vacuum. Although we distinguish between specific heat under constant volume systems or constant pressure systems, the Carnot cycle is neither, so the formula for energy or enthalpy that applies to these systems needs modification, given by the logarithmic functions of volume or pressure used to calculate the Gibbs function. On page 28 of his book, Clausius [7] explained how the higher pressure in a working fluid at a higher temperature could do more work than a fluid at a lower temperature and pressure; Carnot interpreted the same changes in relative volumes in the isothermal Stages 1 and 3, but at different temperatures, as the source of net work because of the variable elasticity of the gas. These different approaches by Clausius and Carnot, completely unknown to one another, confirm how the product of pressure (*p*) and volume (*a*^3^) in a fluid is constant at a given temperature, as a mean value for all material particles. The mean volume *a*^3^ is that occupied by each molecule in a system of volume *V* (*V* = N*a*^3^).
*pa*^3^ = *kT*; *p* = N*kT*(10)

A feature of the virial theorem for free particles in molecular systems is that while kinetic energy obviously changes directly with temperature if volume changes, the configurational potential energy or ergal of the field varies logarithmically with temperature, proportional to *T*ln*T*, it also varies logarithmically with volume or internal pressure [ln(*V*) or ln(1/P)]. Carnot was aware of this in defining his cycle for heat engines [4]. Boltzmann [9] would also perceive this to good purpose in his definition of entropy as a function of *k*ln (*T*^3/2^*V*) as well as by the statistical distributions celebrated on his tombstone [*S* = *k*ln*W*]. Given entropy as defined by Clausius is a dimensionless number, a ratio of extensive to intensive energy [*S*, J K^−1^] it is able to fulfil various statistical functions as in information theory. This paper will show that these complementary results from the virial theorem regarding forms of thermal energy also form the basis of the logarithmic relationship of action to entropy and the conversion of entropy to entropic forms of energy denoted by the scaling factor, absolute temperature [K]. 

The concept of ergal also has an important role in the reversible nature of heat and work. We can see its significance in the hydrological cycle on the Earth’s surface and in the atmosphere. This comprises an internal ergal driven by the evaporative power of sunlight, water eventually replenished at the surface as rain, having transported vast quantities of thermal energy into the atmosphere. Given that the varying Gibbs energy of the ergal is neglected [4] in favour of the sensible heat content of air, an important source of variation in climate may be ignored. It is unfortunate that Regnault’s careful measurements in the mid-19th century showing that the specific heat (heat capacity) of gases is independent of density was widely misinterpreted as invalidating Carnot’s conclusion regarding logarithmic changes in volume or pressure. Even Clausius [7] failed to perceive that Carnot’s principle of the temperature-dependent elasticity of gases required a varying ‘specific heat’ per molecule for the isothermal stages of Carnot’s heat engine cycle; Clausius unfortunately never had access to Carnot’s original work of 1824 [4].

On the contrary, our analysis has shown [4] that the Gibbs energy does vary with the isothermal changes in gas volume affecting density. The source of this misinterpretation was the fact that Regnault’s measurements [10] at one to ten atmospheres pressure were all conducted at constant pressure, lacking processes involving significant change of volume and action state. The marginal change in temperature of one degree Celsius or Kelvin normally used in heat capacity measurements has a negligible effect on logarithmic variation in pressure, within the error of the measurement [4]. However, this change in Gibbs energy can be estimated for the atmosphere having a surface pressure controlled by its gravitational weight. We propose that a significant part of Clausius’ ergal is the work performed on the internal molecular structure required to sustain the pressure of the external work. For reversible systems, these should be equal as shown in Figure 1, although this is not true for irreversible systems where the external work performed will be less than the internal work necessary to provide the expansion. 

### 1.4. Clausius’ Entropy and the Ergal 

As defined by Clausius [7,11], entropy can be considered a property of state measuring the “self reservoir of heat required to raise the temperature of a system of ideal gas molecules to the temperature *T*”. Clausius pointed out that the heat indicated by the entropy must include sensible or detectable heat affecting temperature or kinetic energy as well as ‘work-heat’, where this form of heat included in the entropy was actually “nowhere present” [11], effectively abolishing the caloric theory’s axiom of the conservation of heat. Internal heat content and internal and external work were regarded as reversible in nature, subject in systems at equilibrium to conservation in the first law of thermodynamics. The virial theorem validates these changes of state. The theorem was applied [8] in a mechanistic hypothesis of atmospheric warming, establishing a lapse rate in the temperature fall with altitude for dry air of 6.9 K per km as thermal energy is replaced by gravitational work. However, this entropic energy is positive, representing internal work as translational quantum numbers increase with height as energy is reassigned to gravity (*mgh*) that can be considered as an extension of the molecules’ ergal. The use of ergal could be a useful distinction, as entropic energy (*ST*) is a term rarely used, and it allows the corresponding decrease in Gibbs energy within the molecular system to be distinguished from internal or external work such as lifting a weight against gravity or activating a flywheel.

The potential energy of the action field referred to as the ergal could be considered as a complement to work in many chemical or biochemical processes, largely as a result of variations in the molecular bonding energy of reactants and products. The more firmly material particles are bound to each other, the greater the bonding energy that was simultaneously released as quanta into the local field, increasing their kinetic energy and thus lowering their remaining potential energy [−*mv*^2^, J] as the virial theorem predicts. This is obvious in heat exchanges occurring chemically, but we also see binding energy made available in larger-scale processes, such as the cyclical hydrological flows of water under gravity and in the multidimensional air flows of cyclones and anticyclones [4,8]. All such cycles involve periodic local absorptions of heat as work into the action field, which is followed by energy release in coupled processes achieving thermal work-raising temperature, which itself is driven by the impetus of these energy flows as in turbulence. 

These reflections on the different forms of energy providing a causal basis for action called for a simpler method for calculating entropy and free energy [3,4]. The total thermal energy required to sustain a molecular system, termed here the entropic energy (*ST*), can be expressed as the product of its entropy [*S*, J K^−1^] and its absolute temperature (*T*). As defined by Clausius in 1875 [7], molecular entropy measures all the thermal energy needed to heat from zero Kelvin and then to sustain the evolving system, composed of kinetic heat affecting temperature as well as its work of configurational arrangement or conformation, which does not. While Boltzmann’s famous equation for entropy *S* = *k*ln*W*, with *W* a statistic indicating diversity in configurations, is correct under isothermal conditions, it is less than the whole story [3,4], since temperature variesaffecting the diversity. This formula omits the enthalpic component of entropy, which is inherent in the virial theorem as kinetic heat, stating that the long-term average of the potential energy of a system of gravitationally bound particles is twice its average kinetic energy. Ultimately, this principle must explain the symmetry of variations in kinetic energy and radiant field energy as heat becomes work within each system, which is displayed in the higher electronic states of the hydrogen atom as increased entropy. 

## 2. Action Mechanics

An essential feature of action mechanics is the role of action, with dimensions similar to angular momentum [*mrv* J.sec], but a scalar including progressive angular motion [*mrvδф* J.sec]. Although angular momentum is usually designated by the symbol *L* (Moore, 1972) or *J*, we have preferred [1,4] the @ symbol [J.sec] to emphasise its non-vectorial quality, as relative orbital action. The @ symbol is normally considered to indicate a rate and in philosophy to indicate the actual state, which is appropriate for our purpose. This does not require that action indicates circular paths but emphasises that all radial motion is non-linear, except as an abstraction of infinite radius. Relative action defines this property for the trajectories of particles per radian, allowing meaningful comparisons at different radial separations. The key component of mean radial separation (*r*) for N material particles is proportional to the cube root of volume *V*. Thus, *V* is equal to N*a*^3^ or 8N*r*^3^. Although the impulsive momentum δ*mv* (J.sec m^−1^) plays a critical role for conservation of linear momentum and force as the key feature of Hamiltonian theory, in action mechanics, impulse is measured as radial variations in action (δ*mvr*, J.sec) and its rate (*d*@/*dt*) J or Newton-metres as torque, establishing temperature for free molecules. We will apply this concept of radial impulse when we are considering its role in catalysis. Action is well known as the product of energy and time (J.sec), which is consistent with Planck’s quantum of action (*ħ*, J.sec) and with this discussion. 

Apart from radial separation, the other variable component of action by material particles is the mean velocity *v*, which is calculated from the root mean-square speed; this is shown in Table 3 (in Section 2.1) as proportional to the square root of the temperature in Kelvin. Overall, the translational action @_t_ varies as (3*kTI*)^1/2^/z_t_^1/3^ (see Table 3 in Section 2.1). Relative rotational action @_r_ for a diatomic system is determined with the formula (2*kTI*/σ). *I* indicates the respective translational and rotational inertias (*I_t_* = *mr*_t_^2^; *I*_r_ = *m*_1_*m*_2_*r*^2^/(*m*_1_ + *m*_2_)). These forms of relative action can be used to determine the configurational field of energy inferred by Clausius’ virial. Compared to the reductionist factors of temperature, volume, or pressure normally applied in equations for statistical mechanics, molecular action is holistic, combining these terms in a physical property we claim to be realistic and a useful function for developing theory. 

For a mole of diatomic molecules in a canonical ensemble as defined by Gibbs [12], a concise relationship between the total or entropic energy, involving a thermal term (*RT*lne^7/2^) and relative entropic energy for a diatomic molecule or air can be expressed in Equation (11). Note that this approach is only valid for calculations assuming mean values for action and energies for ensembles of molecules reflecting the Maxwell–Boltzmann distribution [13]. For estimating translational action, a correction for the finite difference between the root-mean-square velocity and the actual mean speed of the molecules is also required (Table 3 in Section 2.1), the former being 1.085 times greater [13].
*ST* = *RT*ln[e^7/2+f(*ħ*ω/*kT*)^(3*kTI*_t_)^3/2^(2*kTI*_r_)Q_e_/(*ħ*^5^z_t_σ)] = *RT*ln[e^7/2+f(x)^(@_t_/*ħ*)^3^(@_r_/*ħ*)^2^Q_e_] = *RT*ln[e^7/2+f(x)^n_t_^3^j_r_^2^Q_e_](11)

In Equation (11), the dimensionless ratio *ħω/kT* varying with temperature (*T*) is expressed as x and σ, z_t_ and Q_e_ refer to constants correcting for molecular symmetry (σ,z_t_) or mean velocity (z_t_) or electronic partition (Q_e_), which is only needed at very high temperatures. Rarely at ambient temperatures, some molecules such as oxygen (O_2_) exhibit electronic partitioning, because of electrons in unpaired orbitals, giving three states with minimal differences in energy level, all of which are occupied. This asymmetry requires configurations of oxygen molecules to be considered as three separate species, increasing its translational entropy by *R*ln3 or 9.13 units [(J K^−1^). This requires 2.723 kJ mol^−1^ of extra heat to bring oxygen to 298.15 K from absolute zero [4]. 

Thus, the entropy of diatomic molecules can be functionally partitioned into several contrasting terms related to enthalpy (*kT*lne^7/2^), rotational energy (*kT*ln(@_r_/*ħ*)), vibrational energy (*kT*lne^f(*ħ*ω/*kT*)^)—these three do not change unless the temperature changes—and a translational energy term (*kT*ln(@_t_/*ħ*)^3^) that changes with pressure or concentration as well as temperature. Changes in temperature will cause changes in all four of these partitioned entropy terms. Conversely, if temperature remains constant, only translational entropy changes occur when pressure varies. Much of the confusion regarding entropy in its composite forms results from not understanding this complexity. Often, discussions of statistical variations in entropy are couched solely in terms of changes in physical configuration or conformation, but this is only true for constant temperature systems, existing only rarely.

Ignoring the vibrational component of entropy as relatively small at ambient temperatures and pressures, we can partition the entropic energy *ST* (J) of a mole (ca. 6 × 10^23^) of diatomic molecules into its separate elements of internal energy (*E* = 2.5*RT*, J), enthalpy (*H* = *E* + *RT* = 3.5*RT*, J), Helmholtz (*A*, J), or Gibbs energy (*G*,) [3,4]. It is important to understand that these values are absolute for each molecule, which are expressed per mole by multiplying by Avogadro’s number 6.022 × 10^23^. Since it is possible to calculate these, at least for ideal gases, there is no need to refer reactions to standard states of 1 molal concentration or 1 atmosphere.
*ST* = *RT*ln[e^7/2^(@_t_/*ħ*)^3^(@_r_/*ħ*)^2^] = *RT*ln[e^7/2^(n_t_)^3^(j_r_)^2^]
= 3.5*RT* + *RT*ln[(n_t_)^3^(j_r_)^2^] = *H* − *G*(12)
= 2.5*RT* + *RT*ln[e(n_t_)^3^(j_r_)^2^] = *E* − *A*(13)

In the above equations for entropic energy, n_t_ and j_r_ are mean quantum numbers for translational and rotational action, respectively, normalising action as ratios by reference to Planck’s quantum of action. Their use is justified by Boltzmann’s statistical definition of entropy (*S = k*ln*W*), where *W* is a statistical measure of diversity and for statistical mechanics [4] as Supplement 1 at the MDPI website. 

Gibbs and Helmholtz energies per mole can also be written in their inverted or proper forms as follows. These formulae provide absolute values of these free energies, given that they are zero when the quantum states indicated by n_t_ or j_r_ are minimal near unity.
*G* = *RT*ln[(@_t_/*ħ*)^−3^(@_r_/*ħ*)^−2^] = *RT*ln[(1/n_t_)^3^(1/j_r_)^2^] = −*RT*ln[(n_t_)^3^(j_r_)^2^](14)
*A* = *RT*ln[(@_t_/*ħ*)^−3^(@_r_/*ħ*)^−2^/e] = *RT*ln[(1/n_t_)^3^(1/j_r_)^2^/e] = *G* − *RT*(15)

Gibbs and Helmholtz potentials or free energies commence at zero and decline as negative values as the entropy increases. They indicate the capacity of the molecular system to absorb more thermal energy, either by increases in temperature or volume. They are at their maximum in the most dense cold conditions. Often referred to as free energies, their character is more indicative of being free of field energy and the low state of action and quantum numbers. Every absorption of a quantum of energy involves a decrease in Gibbs or Helmholtz energies and an increase in entropy. This was a key statement of Planck [14] when he introduced the quantum of energy to avoid negative entropies. The Gibbs energy is the appropriate property to use in constant pressure systems where some of the work potential is needed for pressure–volume work mediated by translational kinetic energy. The Helmholtz potential indicates the maximum work potential in a constant volume system. It is perhaps unfortunate that the Gibbs free energy or chemical potential involves an opposite sense to that of potential energy in the first law of thermodynamics, which is the result of a work process akin to the ergal. By contrast, the Gibbs free energy or thermodynamic potential indicates a capacity for work to be done on the system such as for chemical reaction involving the uptake of thermal energy, leading to increased temperature and increased volume. These are also the conditions that have increased relative action as *mrv* and also of entropy. It may have been Clausius’ intention in introducing the ergal to remove this source of confusion. Until this meaning of chemical potential is understood, thermodynamics can be difficult to follow. 

Equations (12) and (13) involving entropy (*S*) and entropic energy (*ST*) can be written in their more usual forms. These are more frequently expressed in their differential forms as Δ*G*, Δ*A*, and Δ*H* to indicate different states or changes of state in chemical reactions. As systems gain and dissipate heat energy, the second law proposes that the Gibbs and Helmholtz energies of systems can only decrease as entropy increases. Nevertheless, it is not true that these values must always decline for a given system as they can be reversed in a refrigerator.
*G* = *H* − *ST**A* = *E* − *ST**H* = *E* + *RT*(16)

We will investigate later in this paper how vibrational and electronic energy states have a role in controlling reaction rates. For chemical reactions where internal energies designated by *E* may change, we also have to consider the resultant change in bonding energy between reactants and products. This energy change in enthalpy (*H*, J) is allocated to the Gibbs energy change for the reaction, which is used to heat or do work on molecules external to the system raising their entropy. For constant temperature, the internal Gibbs energy, equal to the Helmholtz potential for individual molecules of reactants and products, will remain the same. 

If the relative numbers of reactant and product molecules with different bonding energies change, radiant heat will be released, changing the enthalpy. The Gibbs energy also varies as the concentration of the reactant and product molecules changes. In rare cases of spontaneous reaction, the changes in conformational energy between reactants and products are such that the overall enthalpy change in the system is considered positive and heat is extracted from the environment, reducing external entropy. It can be noted that these releases or absorptions of field energy as heat or cold in chemical reactions will be reflected in changes of mass according to Einstein’s *E* = *mc*^2^, but these changes in mass are so small they can be ignored in chemical reactions. 

In this article, action mechanics will be applied to determine chemical reaction rates and conditions for equilibrium in a revision of Eyring’s absolute transition state theory [15,16]. Despite its initial promise for the study of reaction rates from the mid-1930s, this theory has had little development in recent years. This paper aims to provide some novel clarifying tools that may help rejuvenate this area. 

### 2.1. Methods

All thermodynamic calculations on entropic energy in this paper are based on Equation (11). Table 1 shows primary data related to mass, bond lengths, moments of inertia for non-linear molecules such as water, and the vibrational frequencies needed to carry out these calculations. To aid this process, a computer program (Appendix A, Entropy 6/cal) designed to calculate translational, rotational, and vibrational entropies as well as heat capacities for gas molecules in one pass was prepared. Its decision-making capability is indicated in Figure 2, facilitating thermodynamic calculations for all reactants and products needed in this paper. The only limit to the theoretical accuracy of the results is that of the inputs shown in Table 1, although departure from ideality also affects results marginally. The results published for the entropy of atmospheric gases using this program [3] agreed with third law experimental values to four or five significant figures, even for carbon dioxide. 

The average bond enthalpies [17] required for calculations in this paper are shown in Table 2. Where the products contain less bond enthalpy in sum than the reactants, heat is released from the internal action field as a result of the reaction. This decrease in bond enthalpy is calculated as a spontaneous decrease in chemical potential when the reaction is conducted at constant pressure of 1 atmosphere and the heat evolved is transferred to the environment, increasing its entropy.

The equations to be employed using this program are given in Table 3.
entropy-23-01056-t003_Table 3Table 3Equations solved to obtain thermodynamic quantities.PropertyRelevant EquationsEntropic energy*ST* = *RT*ln[Q_e_e^7/2^(3*kTI*/*ħ*)^3^(2*kTI*/*ħ*)^2^/(σz_t_)]; H_2_, O_2_, N_2_, CO_2_*ST* = *RT*ln[Q_e_e^7/2^(@_t_/*ħ*)^3^(@_r_/*ħ*)^2^] Translational action@_t_ = (3*kTI*)^0.5^/z_t_^1/3^; z_t_ = 2^3^∙(1.085)^3^ = 10.2297Rotational action @_r_ = (2*kTI*)^0.5^/σQ_e_Electron multiplicity partitioning translational entropyσ,z_t_Rotational and translational symmetry constantsVibrational actionΣ*k*[*x*/(e*^x^* − 1) − ln(1 − e^−*x*^)]x = *hν*/*kT*Vibrational heat capacityΣ{*kTx*^2^/[2(cosh*x* − 1)]x = *hν*/*kT*Glossary*Action* [@, J.sec]—a scalar or non-directional measure with dimensions [MRVδθ], a finite spatial segment of momentum; action is relativistically invariant.*Angular momentum* [J.sec]—a conservative vector with dimensions [MRV].*Enthalpy* [*H*, J]—consists of internal energy as sensible heat and heat of atmospheric expansion *RT*.*Entropy* [*S*, J K^−1^]—a pure statistic for cumulative sensible and latent heat content per Kelvin or to indicate information.*Ergal* [J]—heat required for internal work of dissociating bound molecules and for raising their quantum state required by the temperature.*Gibbs energy* [*G*, J]—the potential to store thermal energy as work, decreasing in magnitude as this is achieved.*Radial inertia* [MR]—denotes the radial path of matter, the product of momentum by time, a component of action.
Figure 2Flow diagram for computing absolute entropy and Gibbs energy (see Table 3 for relevant algorithms). A fully annotated description of the relevant algorithms and subroutines to compute entropy, Gibbs energy and enthalpy is available as Appendix A or on request to the corresponding author.
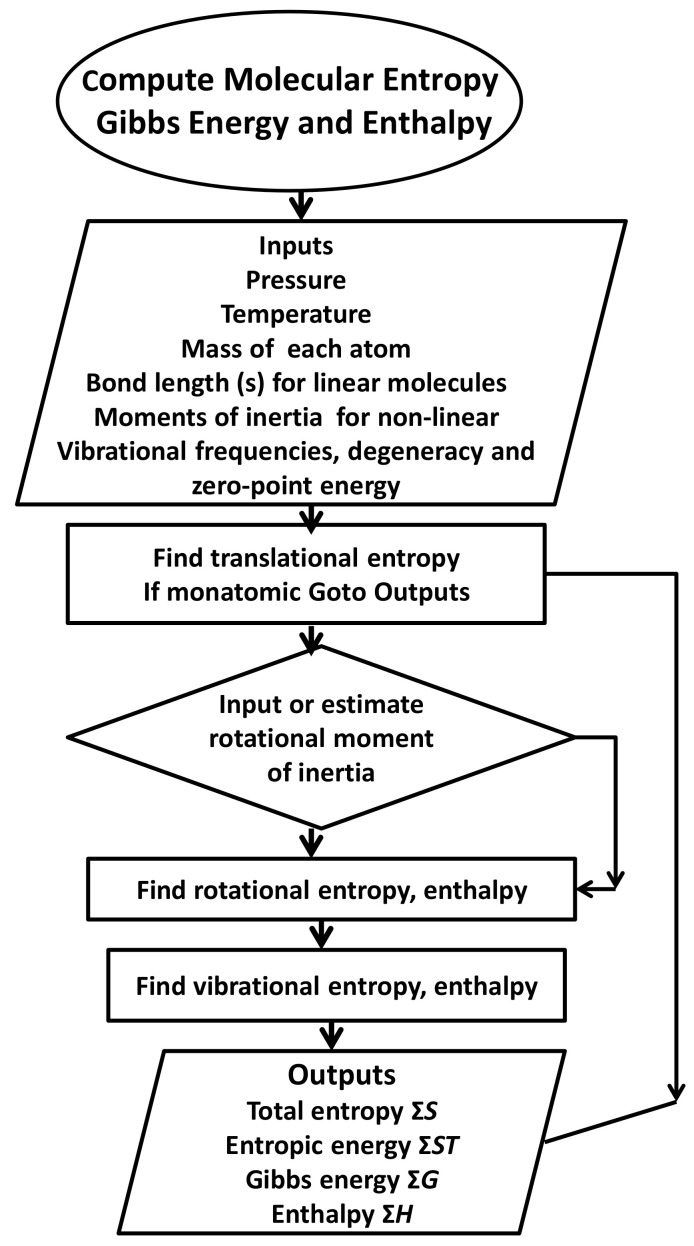


## 3. Activated Transition States

### 3.1. Reaction Kinetics 

At the ambient temperatures of ecosystems, most chemical compounds are stable; atoms are held together by bonding energy that was released to the universal field when the molecules were formed from their constituent primary particles (nucleons, electrons). As a result of an energy barrier substantially exceeding *kT*, their constituent elements such as C, N, P, S, Fe, Ca, Mg, etc. are unavailable for use by living organisms unless mobilised. To mobilise or release these elements in natural cycles, it is essential to provide the required activation energies to break bonds, allowing transition states to be achieved, from which desirable products may be synthesised. Such activated transition states are characterised by higher vibrational and translational action (and lower concentration) than the more stable ground states in which these compounds are normally found. 

Higher action states can be generated in the following transitions:(i)Absorption of energy quanta or photons, which activate the electrons in the absorbing molecules (N) to excited states (N*), present in concentration ratios given by the Boltzmann distribution, where ε is the difference in energy between the ground and excited levels.
N*/N_0_ = e^−*ε*/*kT*^(17)(ii)The larger the quantum of activation energy required for activation (*ε* = *hν*), the lower the concentration of the activated state N*. In general, the rate of the chemical reaction resulting from activation is dictated by the concentration or frequency of occurrence of the activated transition state, a state of lower free energy and higher entropic energy. According to Planck [14], any such activation as the result of the absorption of quanta increases the entropy, decreasing the Gibbs energy.(iii)The absorption of photons in photosynthesis [1] raising electrons in the two photosystems of green plants (*PSII* and *PSI*) to higher action states of greater mobility is one important example of this process. The absorption of resonant infrared photons exciting vibrations in gaseous molecules such as those shown in Table 2, emitted from the earth’s surface by polyatomic gas molecules such as H_2_O, CO_2_, CH_4_, or N_2_O, is another.(iv)As indicated by Equation (17), raising the temperature so that the ratio *ћ*ω/*kT* declines provides another means of increasing the proportion of molecules in the activated state needed to allow a chemical transition to take place. This requirement explains why life only exists within an optimal temperature range. If conditions are too cold, below freezing, biological molecules will be too immobile for life. If too hot, the lifetime of biological bondings will be too short for stable organisms to occur.(v)The most common means employed by living organisms to mobilise chemical species is by providing catalytic agents or coupling mechanisms, lowering the activation energy for reaction, or by providing a forceful mechanism to achieve it; thus, this raises the frequency or likelihood of activated transition state species and increasing the reaction rate as discussed below. This lowers the size of the quantum of energy *h*ν needed to reach an activated state allowing transition.(vi)This process of reducing the magnitude of the activation energy (actually reducing the negative Gibbs energy barrier) is known as catalysis, and the biological catalysts involved are colloidal proteins called enzymes.

Evolving these specific coupling mechanisms or enzymes, which are able to direct metabolism by lowering activation energies, is one of the main tasks of the genes of living organisms [1]. Thus, enzymes as molecular products of specific and purposeful genetic information destabilise all kinds of stable structures, mobilising their constituents and setting up molecular flows into alternative products.

However, it is important to recognise that the catalysts in living organisms known as enzymes can also act to immobilise or fix simple compounds, such as atmospheric CO_2_ and N_2_ in the processes of photosynthesis and biological nitrogen fixation; these are major processes on Earth on which life utterly depends. 

### 3.2. Action Revision of Eyring’s Absolute Transition State Theory 

In the molecular reaction with activated transition states of reactant and product (*A**, *P**), formed from standard states of reactant (*A*) and product (*P*), 1 molal or 1 atm pressure,
(18)Ao ↔ A* ↔ P* ↔ Po,
the Eyring transition rate theory sought to generate a rate constant for the forward reaction. The model here uses a single molecule of reactant and product in the standard state to simplify the description. However, *A* and *P* might be molecular isomers in different conformations, having changed bond lengths affecting moments of rotational inertia (*I*_r_) and vibrational frequencies.

The rate constant *k_f_* for the forward reaction was proposed to be a function involving a natural vibrational frequency of chemical bonding around 6.25 × 10^12^ at 300 K, from the energy-action quotient (*kT/h*) of Boltzmann’s constant (*k*) and Planck’s constant for the action of all quanta (*h*), ignoring other possible factors such as the transmission rate, which is assumed here to be 1—that is, all molecules reaching the transition state are on a trajectory of conversion to product. Then, a quasi-equilibrium between the reactant *A* said to occur under standard conditions and the activated transition state *A** leads to Equation (19).
(19)$$k_{f}=\frac{kT}{h}K_{f}*,\;\mathrm{where}\;K_{f}*=[A*]/[A^{o}]$$

By assuming that both *A* and *A** are at unit concentration [16,18], it is said to follow that *e*^−Δ*G**o/*RT*^ is equal to *K_f_*. Thus,
(20)$$k_{f}=\frac{kT}{h}e^{-\Delta G^{*o}{}_{f}/RT}$$

Since *e*^−Δ*G*of*/*RT*^ = *K**
(21)$$k_{f}[A^{o}]=\frac{kT}{h}[A*]$$

So, the overall rate of the reaction producing *P* from *A* is proposed to depend simply on the concentration of a transition state [*A**]. This is not disputed.

Eyring’s absolute transition state theory assumes that the transition intermediate *A** is in equilibrium with *A* in the ground state [15,16,18]. Then, the standard free energy change for this equilibrium is given in the following equation.
Δ*G^o^** = −*RT*ln*K**
*k_f_* = *kT/h e*^−Δ*Go*/RT*^ = *kT/h e*^ΔS*o*/R*^*e*^−ΔH*o/RT*^(22)
ln*k_f_* = Δ*S^o^**/*R* − Δ*H^o^**/*RT* + ln *kT/h*(23)

This division into standard entropy (Δ*S^o^**) and enthalpy (Δ*H^o^**) of activation [19] has led to much discussion regarding differential effects of configuration and of energy on providing the activation energy. Frequently, the fact that the *K_f_** is actually equal to *e*^−Δ*Go*/RT*^ for the standard free energy change is ignored, and *e*^−Δ*G*/RT*^, a non-standard relationship not necessarily equal to the equilibrium constant *K* is assumed instead.

In addition, the related Arrhenius relationship has been useful for graphical analysis.
 dln*k_f_*/d*T* = (Δ*H^o^** − *RT*)/*RT*^2^= *E*_a_/*RT*^2^(24)

However, these equations imply that both *A* and *A** are in their standard states of unit concentration or pressure, that is, as *A^o^* and *A^o^**. Logically, this absolute version of the Eyring theory then requires that the resultant equilibrium constant *K** is unity, since it equals [*A^o^**]/[*A^o^*], when Δ*G^o^** must then be zero. It is legitimate to question this assumption in the Eyring theory as a possible flaw in the theory and inconsistent with reality; a unit concentration of the activated transition state as assumed seems impossible, given its extremely highly activated state. Then, the status of Equation (23) and the relevance of the discussion may be uncertain. Many authors [19,20] avoid this flawed reasoning by ignoring the requirement for standard conditions, referring to Δ*G** only, but this must lead to errors.

By contrast, in action mechanics, it is proposed that there is an equilibrium achievable between the states *A* and *A**, with zero Gibbs energy difference between both states as a result. Instead, the equality is actually between a Boltzmann distribution of internal vibrational and electronic states of the relevant bond in *A* determined by its varying bonding energy and the translational quantum state of the activated species, possessing a much longer radial separation, inertia, and action. Thus, a decrease in internal Gibbs or chemical potential caused by activation directed to a bond-changing transition state is equal to that between the ground translational state and the translational state of the activated species. Although ground and activated states have the same mean temperature and kinetic energy, the high inertia of the activated state [m*r] (i.e., its low concentration) requires much greater sustaining field energy. This implies that the number density N* and pressure or concentration of the transition state *A** will depend on the internal vibrational or electronic bonding energy Δ*E*_v_, or Δ*H*_v_.
*AA*_external_ ↔ *A***A* ↔ *AA*_internal_
N*/N^o^ = *K** = *e*^−Δ*E*v/*RT*^(25)

We propose that the relative concentration of the transition state can be directly estimated by substituting the bonding enthalpy Δ*E* or Δ*H* in Equations (17) and (25). Using action mechanics, we can also calculate the change in entropic energy of the conversion of *A* to *A**. At ambient temperatures, changes in vibrational entropy are relatively small and can be ignored. So, for a mole of diatomic molecules, we will have the following difference between activated and standard ground-state translational and rotational entropic energies.
*T*Δ*S = RT*ln[e^7/2^(*@_t_/*ħ*)^3^(*@_r_/*ħ*)^2^] − *RT*ln[e^7/2^(^o^@_t_/*ħ*)^3^(^o^@_r_/*ħ*)^2^](26)

Given that the action varies in the same heat bath, both species have the same temperature, and we can eliminate changes in kinetic enthalpy and rotational energy, neither varying with concentration.
−Δ*G* = *RT*ln[(*@_t_/*ħ*)^3^ − *RT*ln[(^o^@_t_/*ħ*)^3^]= *RT*ln[(*@_t_/^o^@_t_)^3^]= −*RT*ln[(^o^@_t_/*@_t_)^3^]= −*RT*ln[(N*)/(N^o^)](27)

Thus,
N*/N^o^ = *e*^Δ*G*^_t_^/*R*^ = *e*^−Δ*E*v/RT^*= K_f_** = [*A**]/[A_o_]. (28)

In action mechanics, we conclude there is equality between the change in vibrational and electronic energy—Δ*E*v on just breaking a bond and in the difference in translational statistical or Gibbs energy Δ*G* between the ground state molecules and the decrease in Gibbs energy in the transition intermediate. This result is consistent for an equilibrium reaction where Δ*G* for the conversion *A* to *A** is zero, given Δ*G* = Δ*H* − *T*ΔS. 

These results can be interpreted as indicating that the collisional work performed in raising the vibrational energy from the translational energy of a ground state molecule *A* to the transition state *A** is equivalent to the statistical entropic work between the translational states of *A* to that of the transition state *A**. Each molecule of *A** will have much lower concentration or greater separation and action than ground state molecules of *A*, but moving with a greater moment of inertia, colliding with greater and more sustained force. However, this effect will be more than offset by the lower frequency of occurrence of such activated species in uncatalysed systems. These molecular processes can be considered to involve a specific increase in entropy of a molecule of *A* by the absorption of thermal energy from the ground state. A high density of state implies a higher Gibbs energy, with more potential to be activated in collisions. 

For the reverse reaction of conversion of *P* to *A*, which is usually neglected in discussions on transition state theory, we must have an analogous set of relationships. Thus, using the same theory as for the forward reaction, we have Equation (29).
(29)$$k_{r}=\frac{kT}{h}K_{r}*,\;\mathrm{where}\;K_{r}*=[P*]/[P^{o}]$$
(30)$$k_{r}=\frac{kT}{h}e^{\Delta G^{*o}{}_{r}/RT}$$

This implies that the concentration of the activated transition state *P** dictates the rate of the reverse reaction. Then, according to the action mechanics transition state theory, if Δ*G*^o^_r_* is greater than Δ*G*^o^_f_* (i.e., more negative), the rate of the forward reaction will be greater than the rate of the reverse reaction because the concentration of *A** will be greater than the concentration of *P**.

From this action analysis, it is clear that the changes in standard free energy, Δ**G^o^*, which are usually suggested to be involved in Eyring’s theory and equated to Δ**H^o^* − *T*Δ**S^o^* are then consistent with Equation (31).
(31)$$k_{f}=\frac{kT}{h}K_{f}*\;\mathrm{where}\;{K_{f}}^{*}=[A*]/[A_{o}]\;\mathrm{is}\;\mathrm{equivalent}\;\mathrm{to}\;k_{f}=\frac{kT}{h}e^{\Delta G^{*o}{}_{f}/RT}$$

Realistically, a standard or unit concentration transition state cannot be in equilibrium or even in quasi-equilibrium with its ground state at the same unit concentration. In action mechanics, the effective −Δ*G* factor in the reaction must be the Gibbs energy difference for the theoretical transition **A*_f_ ↔ *_r_*P* from forward and reverse directions and not for the transition **A^o^*
↔
*A*. 

Then, taking both forward and reverse reactions, combining Equations (33) and (34), we relate the forward and reverse reaction rates to the overall reaction equilibrium constant *K_eq_*.
(32)$$\frac{k_{f}}{k_{r}}=e^{-(\Delta G^{o}{}_{f}*-\Delta G^{o}{}_{r}*)/RT}=e^{-\Delta G^{o}/RT}=K_{eq}\;=K_{f}*/K_{r}*=[A*][P_{o}]/[A_{o}][P*]=[A*]/[P*]$$
since
∆*G^o^* = −*RT*ln*K_eq_* = ∆*H^o^* − *T*∆*S^o^*.(33)

Thus, with standard concentrations of reactants and products, it is clear from this action revision that the rate of the forward reaction proportional to *A** is greater than the reverse reaction proportional to *P** by a relative factor equal to the equilibrium constant for the reaction.

For the non-standard conditions that usually exist in ecosystems, most reactants and products will be present at concentrations in the *mM* range of less, rather than 1 molal or at 1 atmosphere; however, reaction rates can be predicted by simply multiplying the respective rate constant by the actual concentration or pressure. Thus, the forward reaction rate,
*k_f_* [*A*] = *kT*[*A**]/*h*
and the reverse reaction rate
*k_r_* [*P*] = *kT*[*P**]/*h*
result in the equilibrium constant.
*k_f_*/*k_r_* = *K*_f_*/*K*_r_* = [*A**][*P*]/[*A*][*P**] = *K_eq_*(34)

For non-standard concentrations of ground-state molecules, the concentrations of reactant and product transition state molecules will differ simply in proportion to the relative concentrations of *A* and *P*. Where reactants and products are in standard states of unity, the ratio of the forward and reverse rates will be equal to the equilibrium constant. In the case where the ratio of [*P*] and [*A*] is equal to the equilibrium condition (*K_eq_*), then the ratio of activated states [*A**] and [*P**] will be equal to 1.0, and no chemical work is possible. These conclusions are key to the consideration of equilibria in the case studies examined later in this paper such as the thermal dissociation of hydrogen and water, where reactants and products are held constant in pressure at different temperatures and the negative Gibbs energies of activation are set equal to the standard enthalpy [17] of the following bondings, H-H, O-H, O=O, N≡N and N-H (Figures 5–7).

The choice of the positive sign for the variation in Gibbs energy of activation in Equations (28), (30) and (31) may be confusing. It is to some extent arbitrary although it must suit its mathematical purpose, controlling rate constants correctly. Where an activation process follows a collisional trajectory, the compressive effect on reducing bond length initially raises the particle density and its Gibbs energy becomes less negative in magnitude. However, the elastic reaction from a strongly compressive impulse will increase the bond length and amplitude of the vibration, reducing the density and increasing the activated transition state action to the point where the bond may be broken, with the Gibbs energy change then strongly negative. This logic from action mechanics explains the positive sign for Δ*G* on these equations, whereas the Eyring theory uses a negative sign with positive Gibbs energy changes. 

### 3.3. Radial Action

The feature mentioned above for vibrational excitation was the quasi-equilibrium or equality between the probability of the excited quantum state to the translational action of the excited particle. Excited states expressed translationally operate with greater radial action than the ground state according to the Boltzmann series. The excited translational action corresponds to a reduction in Gibbs energy compared to that of the ground species. Furthermore, the inertial impetus of the excited species moving on a straighter trajectory experiencing greater Brownian mobility has a capability of exerting greater force in collisions and has greater potential to reach a transition state if the collision process favours orbital steering. In action mechanics, this radial inertia (*mr*) is a defining feature of the quantum state.

Table 4 compares nitrogen and carbon dioxide in terms of this radial effect of increasing excitation for the first two vibrational levels. The much lower probability of the vibrationally excited nitrogen states magnifies the radius by more than 100 times. This means the translational Gibbs energy declines much faster for nitrogen than carbon dioxide. Offsetting this, the frequency of such activated species is much lower for nitrogen rather than carbon dioxide, given that a doubling of radius involves an 8-fold increase in the specific volume for molecules (8*r*^3^ = *a*^3^). 

An uncatalysed transition state depends entirely on thermal activation, being the key variable in the scaling factor e^−*ε*/*kT*^. Radial action provides an explanation of why reaction rates may be catalysed by quasi-inert particles, for example in the Haber process reacting N_2_ with H_2_, to be studied later in this paper. Interaction or surface binding of a species such as N_2_ is proposed to increase the frequency of activated species because of the great radial inertia of the catalyst particles. The velocity of the particle varies inversely with the root of the particle mass, so that the overall linear momentum of the catalyst increases proportional to the mass of the particle. Taken with a much smaller frequency of catalyst particles inversely proportional to its specific volume, it is obvious that collisions involving adsorbed species will have a much greater chance of achieving sufficiently reduced Gibbs energy to achieve a transition state. The negative Gibbs energy of the catalytic particle can be transferred into the internal bonding, disrupting electronic bonding during the collision. 

Previously, the focus in catalysis has been on energy surfaces regarded as static over the reaction coordinate which the activating complex traces, without specifying the nature of this process. The idea that enthalpic and entropic forces control the process as discussed in the Eyring theory can be accommodated in the action model of catalysis, but only as increases in the kinetic energy of activated bonds but also with decreases in Gibbs energy contributing the entropic factor. Given that entropy includes components of both enthalpy and negative Gibbs/Helmholtz, referring to the idea of entropic force may be considered as too general a concept. Entropic variation refers to both the internal and external ergal s of molecules, given that bond excitation includes as much increase in potential energy as in kinetic energy.

### 3.4. Chemical Potential and Work

A chemical reaction can be thought of as an expansion process, such as that of a flow through a membrane from one compartment to another.
[*A*]_A_ ⇐ |[*A*]***|** ⇒ [*A*]_P_(35)

The potential work per molecule (*w*) expanded is given below and per mole.
*w = kT*ln([*A*]_P_/[*A*]_A_) (36)
Δ*G* = *RT*ln([*A*]_P_/[*A*]_A_)(37)

If [*A*]_A_ is much greater than [*A*]_P_, then Δ*G* is negative, and the work potential is spontaneous. Similarly, we can take a chemical reaction under standard conditions of 1 molal concentration (or 1 atm pressure for gases).
*A^o^* ↔ *A**↔*P**↔ *P^o^*(38)

Here, we have a transition process or trajectory in which the substrate molecule *A* is equilibrated in an ensemble with activated *A** able to select the spontaneous trajectory as an activated product molecule *P**, which can deactivate to *P* (*A* ⇒ *A** = *P** ⇒ *P*) with no further work required. From transition state theory, an analogous reverse trajectory from *P* to *P** = *A** to *A* must also occur independently, although at a different rate unless the concentrations of *A* and *P* are at equilibrium when [*A**]/[*P**] equals 1.0 and no reaction occurs. This transition between reactant and product may occur in a collision with other molecules, at a catalytic site (as on an enzyme) or at the wall of a containing vessel (Figure 3).

Excess work potential per molecule reacted under standard conditions is given by the difference in chemical potential between the product transition state and the reactant transition state.
Δ*g* = *kT*ln([*P**]/[*P^o^*]) − *kT*ln([*A**]/[*A^o^*])(39)
= *kT*ln([*P**]/[*A**]) + *kT*ln([*P^o^*]/[*A^o^*])= *kT*ln([*P**]/[*A**]) = *kT*ln1/K = −*kT*lnK (40)

Thus, work potential per mole (N) or standard Gibbs energy is given by
Δ*G^o^* = −*RT*lnK (multiplying by N, Avogadro’s No. for a mole).

An important conclusion also made previously in Equation (32) is that for unit standard states of reactant and product (1.0), the ratio of the activated states (*P**)/(*A**) is equal to the equilibrium constant, as shown in Equation (40). In contrast, under non-standard conditions, we find that the work potential per molecule requires two additional expansion or action changing terms to be inserted for the transitions of *A* and *P* from standard (1 molal or 1 atm) to the actual non-standard conditions:  Δ*g* = *kT*ln([*P**]/[*P*]) − *kT*ln([*A**]/[*A*]) + *kT*ln([*P*]/[*P^o^*]) − *kT*ln([*A*]/[*A^o^*])= *kT*ln([*P**][*A*]/[*P*][*A**]) + *kT*ln([*P*]/[*A*])        = −*kT*lnK + *kT*lnq                 (41)
where q is the quotient [*P*]/[*A*]).

Thus, work possible per mole (N) for non-standard conditions
Δ*G* = −*RT*lnK + *RT*lnq                   = *RT*ln(q/K)                    = −*RT*ln([*A**]/[*P**]), just as under standard conditions.(42)

The effective concentrations of the transition states (*A**) and (*P**) under non-standard states will no longer have a ratio equal to the equilibrium constant. We can conclude that the capacity to do work is caused entirely by the inequality of the concentrations of the two transition states generated from A and P, respectively. Obviously, if *RT*lnK = *RT*lnq (i.e., ratio of product to reactant concentrations = K), then Δ*G* is zero, and no work is possible, for we have equilibrium with equal concentrations of the transition state generated from each direction.

On the other hand, if q = 1 as in standard conditions, then Δ*G^o^* = −*RT*lnK, as shown above. It is noteworthy that any case where the ratio of product to reactant concentrations is 1.0 yields the same result, and standard conditions are not strictly required for Δ*G* to equal −*RT*lnK. This means that the transition state concentrations (*A**) and (*P**) will always have the same ratio as K if the ratio is unity, although the actual concentrations of (*A**) and (*P**) and rates of reaction will depend on the absolute concentrations of (*A*) and (*P*). 

Thus, the rates of chemical reactions can be viewed as the result of a set of action processes in which activated transition states are equilibrated with reactants but not equilibrated with products. Then, product formation can occur as a spontaneous expansion of this non-equilibrated transition state product into the product action field. Furthermore, chemical work is possible by coupling such unbalanced action states of reactants and products to other reactions that can be driven up and energy gradient. For example, the phosphorylation of adenosine diphosphate (ADP) to adenosine triphosphate (ATP) can be driven by a coupled pH gradient [1]. 

The potential for chemical systems in the environment to initiate action, whether biological or not, depends on the existence of unbalanced thermodynamic potentials that offer sources of free energy or work potential. For biological systems, the exploitation of these chemical potentials involves the primary coupling of oxidation/reduction reactions to processes of phosphorylation that enables the overall chemical systems to approach equilibrium.

### 3.5. Comparing Eyring’s Theory and Action Mechanics Transition States

The action revision of transition state theory given here differs from the Eyring model in several respects:(i)In action mechanics, the activated transition state *A** more realistically cannot be considered as actually in a standard state. The concentration is considered to be set as a Boltzmann exponential by the magnitude of the bonding energy of the reactant molecules.(ii)While Eyring’s theory usually analyses reaction rate in terms of Gibbs energy differences between the ground state reactant and the activated transition state molecules, in action mechanics, there can be no difference in chemical potential between the two states. This follows because the increased enthalpy and potential energy and vibrational amplitude of molecules in the activated transition state are corresponding to the decreased chemical or Gibbs energy in each activated molecule, according to the quanta required to be absorbed; this decrease in potential is paid for by the potential of the ground-state molecules. This is essential, given the Boltzmann equilibrium between ground and activated states for the ensemble of molecules.(iii)Proponents of Eyring’s transition state theory often fail to consider the reverse reaction as having a significant role in determining reaction rates. This may be valid in reactions proceeding far from equilibrium. However, action mechanics implies that in reactions closer to equilibrium, the chemical potentials for both reaction directions must be considered, given that the net reaction rate depends on the ratio of the transitions states *A** and *P**. Furthermore, the relative rates of forward and reverse reactions must always comply with the overall thermodynamic properties of the reaction, as shown above.

In contrast to Eyring’s absolute transition theory, the action mechanics transition state theory suggests that overall chemical potential changes for reactions can be estimated directly from the variation in bond energies and the changes in entropic energy of reactants and products. Indeed, using action mechanics, it is possible to do direct calculations of all thermodynamic properties from mechanical properties of the molecules. 

## 4. Catalytic Action

This analysis of the transition state has deemed the energy required to reach the activated transition state as equivalent to the bonding enthalpy of particular atoms or groups involved in reactions. If radiant energy or heat equivalent to the bonding energy can be injected into the bond increasing its entropy, the bond may even break. However, except for the weakest bonds, this may mean that without catalysis, the likelihood of achieving a transition state is exceedingly low. The majority of reactions, except those that are diffusion limited not requiring activation above *kT*, may never occur at a significant rate without some form of catalysis. For sufficiently large bonding energies requiring activation, there may be no activated species in a given volume of reaction at a given time. 

The essence of catalysis, still consistent with transition state theory, is that either the activation energy for transition to a product can be achieved by raising the temperature, perhaps locally by the collisional compression of molecules. Thus, near 4000 K, as shown later, the hydrogen molecule is spontaneously 50% dissociated into two hydrogen atoms. Many chemical reactions in industry are conducted at higher temperature to reduce the need for more activation energy from collisions. 

More remarkable is the action of catalysts that can apparently generate states of activation without the need for increased temperature. For example, by rearranging electronic orbitals, the resonant frequency of a molecular bond key to the reaction can be reduced with bond lengthening; thus, the activation energy to the transition state would be less and more readily provided in a collision. As a result, the concentration of the activated transition state would increase exponentially, in proportion to the reduction in the bond enthalpy. An increase by a catalyst in the frequency of occurrence of activated transition states is effectively an increase in its translational chemical potential. In action mechanics, there is also the possibility of a forge-like increase in temperature at the precise site of the bond to be broken. 

### 4.1. Are Colloidal Catalysts Also Inertial Anvils?

Catalysts are invariably colloidal particles, whether chemical or biochemical in nature. The most sophisticated catalysts are the colloidal proteins coded by DNA known as enzymes, with highly specialised functions of great subtlety, such as intra-molecular group transfers by cobamide enzymes [1]. Some enzymes are capable of enhancing the rate of simple reactions such as hydrolysis or decarboxylation by factors as large as 10^17^ [22,23]. Given their ubiquitous effect, we can enquire as to whether binding to colloidal particles has a unique function in catalysis. 

Being bound to the surface of an inert colloidal particle will have an influence on the inertia of a reactant, in effect the particle providing an inertial ‘anvil’ where the reactant molecule may be properly oriented for further reaction. The inertia of the colloidal particle present at relatively low concentrations must have the potential to exert greater force during collisions, given its high momentum, thus increasing the probability of bimolecular reactions and weakening of bonds. Colloidal particles possess relatively high translational entropic energy being more widely separated, with large rotational energy and larger vibrational energy, given the large number of degrees of vibrational freedom. These properties contribute to more substantial inertia (i.e., tendency to continue their current trajectory), which is subject always to the statistical or stochastic character of their Brownian motion. To an extent remaining to be determined, the bound substrate (Figure 4) must share the greater inertia and negative Gibbs energy of the colloidal particle. 

If the properties of the catalytic colloid also have attractive effects for a second reactant, while orienting the molecule (orbital steering, Fersht [24]), the likelihood of reaction must be increased. The decrease in entropic energy of the reactant molecules on binding to a colloidal surface has been regarded as a contribution of increased Gibbs energy to the reactant molecules, providing a measure of this effect. In addition, catalysts such as enzymes may have specific binding interactions that increase the strength of binding, conferring specificity for binding and subsequent reaction. 

### 4.2. Effect of Binding Energy on Catalysis 

Jencks [23] and his younger colleague, Fersht [20,24], provided extensive arguments regarding the effect of substrate binding energy to a catalyst on reaction rate. The binding of a substrate *A* to a colloid involves a significant loss of translational and rotational entropy, effectively increasing the chemical potential of the substrate, which is now more constrained. However, as indicated by the discussion around Equations (25)–(30), entropy is a complex function, and the loss of entropy in one molecular system may involve an increase in the entropy of another. It must also be understood that a larger Gibbs energy indicates a greater potential to absorb radiant energy to remobilise the system while work is being performed. The conversion of heat to work always involves a decrease in Gibbs energy and vice versa. In action mechanics, there is no obvious role proven for the binding energy of substrates. Just how binding energy could be utilized for catalysis is not obvious, except by heating a strategic location.

Figure 4 shows this activation scheme as a diagram indicating the action path from ground-state reactant through transition state to ground-state product. Incidentally, most authors including Jencks and Fersht have suggested that activation to the transition state must involve an increase in the Gibbs energy. However, recall that a high Gibbs energy is synonymous with a low quantum state, as shown in Equation (14). From the virial theorem [8], internally activated molecules with either increased electronic or vibrational energy must both have increased potential energy, entropy, and diminished Gibbs energy [25]. The increased potential energy of electronic orbitals requires a quantum of radiant energy *hν* absorbed simultaneously with diminished kinetic energy of the same magnitude as the quantum; so the electronic potential energy increases with twice the magnitude of the absorbed radiant energy. A robust collision or compression of a substrate may be involved in the process of catalysis generating repulsive potential energy, the actual achievement of the transition state on rebound with a virial process of absorption of quanta must involve a decrease in internal Gibbs energy. Bound substrates are constrained on trajectories of greater translational action and entropy compared to the unbound state, contributing to the transition state. 

Transition state theory has often focused on potential surfaces that are effectively the same as Gibbs energy profiles, as indicated by the steps shown in Figure 4. The occurrence of both forward and reverse reactions in the figure is not meant to infer that they occur simultaneously at the same site. Far more likely they are separated on separate colloids, reflecting the probability of a collision sufficient to excite the transition state. The lifetime of such transient states is also likely to be short, which is consistent with collision times. Pechukas and McLafferty [26] considered collinear atom–diatom forms of *A* and *P*, proposing a Hamiltonian analysis in which with proper alignment on a saddle point falling away in both directions, proposing that the transition state energy achieved could be exact in value. In future work, we intend to extend the simple model given in Figure 4 to reactions transferring atoms from one substrate to another, which would occur on the same surface (*AA* + *P* > *A* + *AP*). This would only require the facilitation of well-aimed trajectories expressed statistically, either in solution or on colloidal surfaces. The extent to which a binding colloid orients a reactant molecule, perhaps by embedding a diatom *AA*, is more likely to be critical than an exact surface function if the catalysis is mainly from stress caused in collisions. Such tolerance has the advantage of keeping the catalysis model simple, not requiring supercomputing.

The absorption of infrared quanta *hv* increases both the kinetic and potential energy of the vibration by *hv*/2. The Gibbs or Helmholtz energies of the activated molecules must decrease internally, becoming more negative as quanta are absorbed. The greater the activation, the lower the internal Gibbs energy. The translational entropy will become larger as the activation energy is accepted on the catalyst. Thus, the rate equation can be written in terms of translational Gibbs energy.
*k_f_* = *kT*/*h*e^Δ*G*t/*RT*^(43)

In effect, Δ*G*_t_ becomes more negative as the internal activation energy to achieve transition is reduced by catalysis. 

### 4.3. Effect of Transition State Complementarity 

Binding energy is maximized when all binding groups on the substrate or reactant are complementary to binding sites on the enzyme. Haldane and Pauling had suggested it would be catalytically advantageous for the enzyme to be complementary to the activated transition state of the substrate (*E* + *S** ↔ *ES**) rather than the original ground-state molecule *S* before binding. In such cases, the initial binding of the substrate to the enzyme (*E* + *S* ↔ *ES*) may be relatively loose and sufficient to immobilise the substrate and to effectively increase the concentration of other reactants where more than one is involved.

Fersht [22] more recently explained that the full binding energy is realised only when the enzyme is complementary in structure to the transition state. Under these conditions, the initial binding of the substrate to the enzyme indicated by the raised Michaelis *K*_a_ value is less strong. An enzyme with a high *K_m_* value is generally regarded as having a higher catalytic rate. For example, glutamate dehydrogenase (GDH) from lupin nodules with a *K_m_* for ammonia of about 70 mM [27] has a turnover number (molecules of product per molecule of enzyme per sec) some 50 times higher than glutamine synthetase *K_m_* for ammonia 12.5 μM [28] from the same source—an enzyme regarded as having a scavenging role for ammonia that can be poisonous if its concentration is excessive. 

The exact mechanism of the much lower turnover time of glutamate dehydrogenase compared to glutamine synthetase is not fully understood. However, the loose primary binding of ammonia to GDH and high Michaelis *K_m_* requiring a higher concentration to achieve maximum velocity may allow the process of conversion to the transition state to be more firmly bound with the binding energy contributing to easier activation and a high *k*_cat_. It is suggested that these new viewpoints of action mechanics and radial inertia by large colloids offer opportunities for new theoretical developments, not being pursued here. We suggest that the combination of a relatively small colloid in GDH with large radial inertia (*mr*) and a high *K_m_* value with a high probability of binding substrate molecules though loosely compared with the much larger GS molecules at a much higher concentration (ca. 100 times) and low radial inertia and very low *K_m_* with tight binding is consistent with the collision hypothesis we have advanced using action mechanics.

## 5. Steady States and Equilibrium 

We have prepared a single program capable of calculating all results needed in this paper, based on their molecular properties given in Table 1. This program is attached to this paper as Appendix A.

Chemical equilibrium occurs when both systems of molecules in a reaction have the same Gibbs energy. That requires that the action or quantum fields are equal and the concentrations of activated transition states for forward and reverse reactions are also equal, as discussed in Section 3. Gibbs energy can be calculated directly based on Equations (12) and (14). In varying natural systems, equilibrium is rare, and the occurrence of steady states with far from equilibrium concentrations are typical. This has the advantage of maintaining high rates of product formation, provided reactants are replenished. Reactions at equilibrium are more likely to occur in isolated locations or backwaters. 

### Statistical and Action Mechanics

*K_f_** for the activation equilibrium a*A* + b*B* = [**AB*] is given in statistical mechanics as the following Equation (44). The Q factors represent canonical partition factors, which are sometimes rendered as Z values [15].
*K_f_** = [(Q*/*V*)/(Q_A_/*V*)^a^(Q_B_/*V*)^b^] e^−Δ*E*o/*RT*^(44)

Here, *V* is taken as the system volume occupied by the molecules; Δ*E*o represents the difference in bonding energy between reactants and product at absolute zero. These enthalpy values must be corrected at temperatures above zero Kelvin. 

The Q values refer to the overall product of the electronic, translational, rotational, and vibrational partition functions of the reactants. These have been shown [3,4] to be effectively action ratios, indicating quantum states of the reactants and are given in their classical form following: translational partition function, *Q*_t_ = (2π*mkT*/*h*^2^)^3/2^*V*; rotational partition function (linear molecule), *Q*_r_ = 8π*IkT*/*h*^2^; rotational partition function (non-linear molecule), *Q*_r_ = π^2^(8π^3^*I_A_I_B_I_C_*)^1/2^(*kT*/*h*^2^)^3/2^; vibrational partition function (polyatomic molecules), *Q*_vi_ = П_i_ [1 − exp^−*hν*i/*kT*^]^−1^, where П_i_ indicates a product of *i* separate functions, one for each mode of vibration. At normal temperatures, the electronic function Q_e_ can be ignored with a small number of exceptions (e.g., paramagnetic O_2_). 

In action thermodynamics and mechanics [1,4] for the gaseous reaction *A* ↔ *P*, assuming no role for excited vibrational states or electronic states (Q_e_), we can express reactant *A* entropic energy as follows.
^A^*sT* = *kT*ln[e^7/2^(^A^@_t_/*ħ*)^3^(^A^@_r_/*ħ*)^2^]
^A^*sT* = *kT*ln[e^7/2A^(n_t_^3^)(^A^j_r_^2^)](45)

A similar equation can be written for the mean value of entropic energy for diatomic product molecules, where n_t_ and j_r_ are translational and rotational quantum numbers.
^P^*sT* = *kT*ln[e^7/2^(^P^@_t_/*ħ*)^3^(^P^@_r_/*ħ*)^2^]
^P^*sT* = *kT*ln[e^7/2P^n_t_^3P^j_r_^2^](46)

These are easily adjusted for monatomic and polyatomic molecules, as is demonstrated later.

Apart from the exponential term (e^7/2^), the translational and rotational quantum numbers relate to the work term that is associated with changing internal Gibbs energy. As explained in the Introduction, Clausius referred to the internal work done in the system as the ergal, effectively the negative of the internal Gibbs energy related to the configurational entropy term; initially, he used the term in the opposite sense, but it is more convenient as he eventually decided to apply it positively to increasing field energy content as the Gibbs energy becomes more negative. Unfortunately, the term was apparently considered unnecessary when the kinetic work associated with enthalpy (*h*) took precedence. The enthalpic term (*kT*lne^7/2^ =3.5*k = h*) will vary if there is a change in the molecular bonding, the numbers of molecules or the degrees of freedom of motion possessed by the molecule. 

If a single molecule of reactant *A* becomes product *P*, the system’s enthalpy may change, even cooling the system, but if new bonds are formed involving electronic and vibrational rearrangements, a negative enthalpy change (δ*h*) may occur where heat or work is exported to the exterior of the system. Under these conditions, where heat is exported to the exterior, an additional negative change in Gibbs energy occurs. Overall, this is expressed in the work equation for an average molecule following, for an isothermal system.
Δ*g* = δ*h* − δ*sT*(47)

We might write this work process entirely in potential terms, simply by specifying the location of the work (negative δ*g*) in a subscript.
Δ*g*_total_ = δ*g*_external_ + δ*g*_internal_(48)

Spontaneity is shown as a negative δ*g*, either by export of heat from excess bonding energy to the exterior, or as an increase in entropy in the ergal (non-enthalpic δ*sT*). Spontaneity can still exist, though rarely, when heat is consumed in internal bond rearrangements or elimination and δ*h* is positive, cooling the exterior. In this case, the magnitude of the decrease in the ergal, indicating that a smaller energy field is required to sustain the molecular configuration, is decisive. 

The entropic energy of the product will be as given in Equation (49), with changes in translational and rotational action as the only significant variables. For example, if *P* is more highly concentrated or at a higher pressure than *A*, then its translational entropy and energy will be less and there will need to be compensating increases in either the rotational action and entropy or the internal energy of bonding. That is, either the rotational moment of inertia has increased or the bonding energy could be less in *P* than *A*. Increases in either of these properties indicates that more heat is needed to do the internal work of sustaining the physical configuration of the molecule *P* than *A* as either rotation or vibration at the same temperature *T*. 

Here, the bonding energy is given as −*H*, which is considered as a loss of internal entropy as the heat of bonding is lost from the molecular system to the environment. We use enthalpy *H* rather than energy *E* in a constant pressure system as we assume the system is free to expand doing work of lifting the atmosphere. Then, we have the following difference in entropic energy between the reactant and product.
*T^P−A^*Δ*S = RT*ln[(^P^@_t_/^A^@_t_)^3^(^P^@_r_/^A^@_r_)^2^/^P−A^e^ΔH/*RT*^](49)

Equation (49) can be compared to Equation (44), correcting the Q partition values to relative actions similar to Equation (11), as we have described elsewhere [3,4]. Since both molecular systems are at the same temperature, the e^7/2^ enthalpy term is eliminated, as well as the symmetry constants for rotation σ and translation z_t_. Then, we have the following relationship.
*T*Δ*S* = *RT*ln[(^P^@_t_/^A^@_t_)^3^(^P^@_r_/^A^@_r_)^2^] + Δ*H* = −Δ*G* + Δ*H*(50)

When there is no change in the number of molecules, the variation in total kinetic energy is minimal, and the enthalpy change is derived only from differences in the bonding energy. 

When we consider the familiar Gibbs equation for Equation (50) relating Gibbs energy change (Δ*G*) to enthalpy change (Δ*H*) and entropic energy change (*T*Δ*S*), we need to be clear as to the meaning of these terms and where they apply. The *T*Δ*S* term refers strictly to the difference in entropy at *T* between the system’s chemical products and reactants, as depicted in Equation (49). However, the Δ*G* and Δ*H* terms refer further afield to changes both within the reacting chemical system and external to the system, apparently under constant environmental conditions, as given in all standard texts on thermodynamics.
Δ*G* = Δ*H* − *T*Δ*S*

However, this constancy is rarely true. By convention, if Δ*H* is negative, heat or work is transferred to the environment, and both entropy-increasing processes occur externally. As a result of its great thermal mass, the Earth’s surface environment is considered to be constant in temperature, but in fact, all heat-producing reactions will heat the environment—though usually ever so slightly. Furthermore, if heat or pressure-volume work is transferred to the environment and gravitational or other shaft work is done, Δ*G* will be negative. So, Δ*G* refers to entropy changes both within and without the molecular system. That means molecules other than those in the reacting system are heated or have work performed on them, but they are not represented as such in the reaction. An important advantage of action mechanics is that its more realistic nature allows a clearer viewpoint regarding mechanisms of energy transfers as heat or work. 

As stated in Methods (Section 2.1), provided molecular properties such as atomic masses, bond lengths, moments of inertia, bonding energies, pressure, and temperature are known, the entropy, the enthalpy, and Gibbs energy changes can easily be calculated from first principles with no need to refer to standard tables of reaction. Where there are significant changes in vibrational energy between reactants and products, this must also be taken into account, and this will be illustrated with examples following. At equilibrium, Δ*G* is zero and Δ*H* = *T*Δ*S*; under standard conditions of 1 atm pressure or 1 molal concentration, Δ*G^o^* is equal to −*RT*lnK_eq_ as discussed above. From the success of action thermodynamics at ambient temperatures where vibration is usually unimportant, we must conclude that there is a necessary relationship between the quantum number ratios of translational and rotational action only under adiabatic or isentropic conditions.
[(^P^@_t_/^A^@_t_)^3^(^P^@_r_/^A^@_r_)^2^] = 1 or (^P^@_t_/^A^@_t_)^3^ = (^A^@_r_/^P^@_r_)^2^(51)

Note that under isothermal conditions (constant temperature), the rotational quantum number for a reactant or product remains the same irrespective of concentration or pressure, while the system behaves ideally. Only the translational quantum number is subject to change with concentration, thus affecting the chemical potential. The exact form of the equilibrium relationship will depend on particular molecular properties.

## 6. Illustrative Case Studies

Several important reactions of topical interest to industry are examined here using action mechanics. These include reactions that are effectively irreversible at ambient temperatures such as the thermal dissociation water and hydrogen gas and another that is more poised, the Haber–Bosch process for fixing atmospheric nitrogen (N_2_). In the following reactions, all involving hydrogen molecules as reactants, entropies and enthalpies are calculated at different temperatures using a program requiring only inputs given in Table 1, with Gibbs energy changes calculated by difference, to establish the temperatures where equilibrium is closest to achievement. This analysis allows choice of temperature ranges to discover the equilibrium point as well as to establish temperatures ensuring complete lysis of molecules such as hydrogen, water, and ammonia. Note that all estimations are conducted with reactants and products at a standard pressure of 1 atm, without corrections for non-ideality. The Δ*G* values estimated indicate the work possible maintaining the stated conditions constant, which is a situation that may occur as in the steady state, where reactants are replenished and products are removed continuously. This is similar to the work possible from expansion of a chemical species through a membrane, as described in Equation (35) and those following. Many life processes operate in such anaplerotic cycles, replenishing reactants as products are excreted. 

### 6.1. Dissociation of H_2_ to 2H

This reaction lysing molecules of hydrogen gas (H_2_) to two hydrogen atoms at 1000 K under standard conditions is illustrated in Figure 5, showing entropic energies (*ST*), enthalpy (*H*), and standard Gibbs energy changes during the reaction, assuming each gas has one atmosphere pressure. Table 5 gives the thermodynamic changes in *ST*, *H*, and *G*, showing that under 1 atmosphere pressure of reactant and product, equilibrium is achieved at just under 4000 K. Thus, at 6000 K, we can predict that the surface of the Sun consists of separated hydrogen atoms and not hydrogen molecules. The change in enthalpy is partly affected by the change in the kinetic energy of one molecule versus two, but it is mainly a result of the diminution in bonding energy in this conversion. The thermodynamics indicates that the reaction is strongly in favour of formation of the diatomic H_2_ at the temperature of 1000 K, given that the changes in entropic energy and enthalpy are so far from balance; the action field contains far too little energy to sustain hydrogen atoms. According to conclusions made in the discussion on transition state theory (see Section 3.2), the relative concentration of transition species to ground-state species at 1000 K without catalysis (N*/N_o_ = e^−Δ*H*/*RT*^) would be 1.67 × 10^−23^, about four *H_2_ for each mole. By comparison at this temperature, every atom of hydrogen would be activated for transition, yielding relative rates of reaction of 6 × 10^22^. 

From Table 5, we can observe that under standard conditions of 1 atmosphere pressure, the reaction
H_2_ ↔ 2H(52)
is at equilibrium with *K*_p_ or the ratio of pressures (H)^2^/(H_2_) equal to 1.0 at just over 4000 K, with a standard Gibbs energy change of −2 kJ per mole. At 298 K, the reaction spontaneously forms H_2_ releasing energy from the field or chemical work as heat (ca. 436 kJ per mole), whereas at temperature greater than 4000 K, the reaction is increasingly displaced towards dissociation into H atoms, consuming heat from the field needed to sustain the translational entropic energy of the separated atoms. Thus, on stars such as the Sun, the spectrum observed from Earth is for translational or electronic states of H as a gas and no spectrum for H_2_ is observed. Does this mean that in extremely cold conditions in space, we will expect to see hydrogen as the diatomic gas? Using the action methodology given here, we can easily calculate this by running the program at very low pressure and temperature, only changing these environmental parameters.

In Table 6, the ergal for hydrogen of *kT*ln[(n_t_)^3^(j_r_)^2^] is partitioned into translational and rotational work terms and also into mean values for virtual quanta, using the quantum numbers n_t_ and j_r_ derived in calculating the action and the entropy. It is noteworthy that as the temperature rises and the molecular number density declines maintaining constant pressure of 1 atmosphere, the size of the conjugate translational and rotational quanta decline. Even though the translational ergal for by hydrogen atoms and hydrogen molecules increases more than ten-fold between 1000 and 9000 K, the mean size of the quanta sustaining the system only doubles. This reflects the relationship that the ergal varies proportional to *T*ln*T*.

Surprisingly to the authors, the dissociation of hydrogen has received relatively little experimental interest, despite its current relevance for green energy. Langmuir et al. [29,30] conducted excellent studies on the lysis of various gases by exposure to tungsten filaments as heat sources, using heat conduction as a measure of dissociation of the molecules. The Nobel laureate Giauque [31] studied the thermodynamics of this dissociation, obtaining a very similar result for Δ*G*/*T* near 4000 K. Since the maximum temperature possible was just over 3000 K with 68% dissociation, tungsten filaments melted at higher temperature so Langmuir was unable to achieve the temperature of 4000. 

The theoretical pressure of hydrogen and the resultant reduction potential available in ecosystems has been made a special study by the senior author [1,22]. It was recommended by Manning Clark, inventor of the Clark electrode, early in the 20th century to use the negative logarithm to base 10 of the H_2_ pressure instead of electrochemical reduction potential. He called this the rH value, by analogy with pH for acidity. It has the advantage of reporting the actual hydrogen pressure that a chemical system can sustain and is highly appropriate for environmental chemistry. In general terms, the more negative the rH value, the greater the potential for biological action and life systems. For the data in Table 5, the rH uncorrected for temperature is zero. 

### 6.2. Association and Dissociation of Water

The dissociation of water into hydrogen and oxygen is also a reaction of much interest. The senior author in the 1960s once used electrochemical dissociation to prepare oxygen free of nitrogen gas for labelling experiments, given that the hospital oxygen then available contained significant nitrogen. However, prospects of a convenient source of renewable energy using sunlight provides renewed interest in achieving significant breakdown of abundant water using catalysis.

Figure 6 shows reaction energy changes for standard concentrations for hydrogen, oxygen, and water as reactants and product at 298.15 K, which is a reaction that is essentially irreversible under ambient conditions.

Table 7 shows that significant dissociation is achieved only above 4000 K, showing how stable the water molecule is. It is even more stable than hydrogen gas, which requires a slightly lower temperature to dissociate (Table 5). In order to calculate the Gibbs energy change, it is necessary to adjust the change in enthalpy change from that at absolute zero to the temperature of the reaction. The variation in enthalpy change with temperature is given in Table 7, taking into account the numbers of molecules of reactants and products, being three and two, respectively. Table 8 gives the results obtained by the program for calculation of the reaction enthalpy, varying with temperature. 

In Table 7, it was assumed that hydrogen and oxygen molecules remain undissociated, even at 10,000 K. However, this could only be momentarily true on rapid mixing as atomic oxygen and hydrogen will soon be present in the mixtures increasingly as temperature is raised. The same approach can be applied to the mixture of water, hydrogen, and oxygen molecules and hydrogen and oxygen atoms at any temperature. Solving this for equilibrium requires that all reactions compute to a Gibbs energy change of zero. This cannot be achieved for all gases to be at unit activity of 1 atm. Using action mechanics, it is possible to estimate the relative pressures of water, hydrogen, oxygen molecules, and hydrogen and oxygen atoms at any given temperatures. 

The data in Table 7 suggesting that reaction Δ*G* is zero and that *K*= 1.0 with equal pressures of water vapor, hydrogen, and oxygen near 4500 K is clearly incorrect. We have already shown hydrogen molecules are dissociated 50% into atoms at 4000 K (Table 6) and Johnston and Walker [32] using thermodynamics showed that oxygen molecules would be 50% dissociated into oxygen atoms at 3850 K. For that reason, these data should only be regarded of illustrative of the approach, which can easily be extended to include these secondary dissociations. The principle that all chemical potentials should be equal at equilibrium must be obeyed and can easily be employed in calculations. That can be considered as future work not required here. 

The thermal dissociation of water studied here also has a biological analogy in the photosynthesis of water with carbon dioxide [1] at environmental temperatures. This also involves the splitting of water molecules into oxygen molecules and four reducing equivalents that are equivalent to hydrogen atoms, although these are effectively dissociated as protons and electrons. The extremely high temperatures needed to lyse water thermally contrasts with the absorption of sunlight at ambient temperatures. However, it is important to realise that solar energy is dispatched in quantum packages resonant with atoms at a much higher surface temperature of 5900 K. It is the great virtue of the photosynthetic apparatus that it can provide a capturing mechanism for these quanta that is completely inelastic and use this thermal energy of the Sun’s ergal to do chemical work on Earth. 

### 6.3. The Haber–Bosch Reaction 

Unlike the reactions for the dissociation of hydrogen and water, the fixation of nitrogen in the Haber process is well balanced. As a heat-evolving reaction, it is a reversible process and can theoretically be used to store heat by a dissociation of ammonia (Figure 7).

This chemical reaction is currently the most significant industrial synthesis on Earth. Unlike the hydrolysis of water, the Haber process can be regarded as a reversible reaction given its low Δ*G* value. Indeed, at 450 K, its standard Gibbs energy is zero (Table 9), and *K*_p_ is equal to 1.0. At standard conditions, 93 kJ of energy per mole of ammonia formed is evolved, which is a heat release that increases with increasing temperature, given that four molecules of hydrogen and nitrogen are reduced to two of ammonia. Changes in the kinetic enthalpy with temperature for the Haber process is given in Table 10. It has been proposed and demonstrated [33] that the Haber reaction could be used to provide renewable electrical power from sunlight with no need for battery storage. This is possible by using solar power to provide heat to decompose NH_3_ into N_2_ and H_2_ in the daytime and to run the reverse reaction releasing heat at night. Steam generation continuously can produce electricity by traditional turbines. The process could be integrated with industrial chemistry, if augmented with renewable photosynthate in agroforestry [34] or with fossil fuels.

Table 11 uses data for entropy change including vibrational entropy for ammonia and also increased kinetic heat capacity, accepting that mean vibrational energy is half kinetic and half potential energy. These results are significantly different at higher temperatures than those performed neglecting vibrational entropy and should be preferred in establishing the temperature of equilibrium for gases all at 1 atm pressure. Given the large bonding energies involved in these reactants and products, the concentrations of activated transition state molecules are extremely low. Indeed, it is highly improbable that any are present in a reaction vessel at any instant in time. These reactions almost certainly require catalysis on particulate matter of iron or other catalytic surfaces that act to reduce the activation energy with a corresponding increase in the concentration of the transition state *A** (and equally of *P**) [20,21]. A catalytic surface reducing the activation energy will increase the concentration of the transition state from both directions equally. Once the reaction is ignited, the local generation of heat and temperature rise may further catalyse the reaction to an explosive extent. Although catalysts may speed up the achievement of equilibrium and are essential for the process, it must be remembered that they do not change the position of equilibrium.

The actual molecular mechanisms of catalysis will not be considered in any detail in this paper. Suffice it to say that a reactant bound to the surface of a colloidal particle will experience much greater inertial forces during collisions, given the large translational moments of inertia of colloidal particles as a result of their greater mass and greater inertial radius, given their lower concentration. This inertial effect is sustained by the field energy associated with the higher configurational entropic energy of such particles, but the reactant *A* itself is effectively frozen in position on the surface of such colloids, effectively increasing its translational Gibbs energy.

To establish the transition state in either direction, it is assumed that without catalysis, a total activation energy of 2253 kJ per mole (equal to the bonding energy of the atoms) of nitrogen for both reactants is required. For the reverse reaction, 2346 kJ per mole (the bonding energy of the hydrogen atoms in ammonia) is required, so that the concentration of the transition state in the forward reaction is equivalent to 93 kJ per mole, which is more favourable. This determines that ammonia will be formed at a much greater rate than it decomposes. The entropic energy for reactants and product is easily determined, and Δ*G* is determined by difference, as shown in Table 9. In future work, we will apply action mechanics to the catalytic process, testing the hypotheses advanced in this article.

## 7. Conclusions

We have shown in this article how action mechanics can provide a convenient path to practical and theoretical thermodynamics. As a realistic method derived from combining statistical mechanics and quantum theory, we suggest it has the potential to rejuvenate the application of thermodynamics in general sciences from chemistry and biology to environment, which are areas now lacking its guidance. We recommend that action mechanics be considered as a means of instruction and investigation for senior students. Specifically, we claim advances have been made in this article including the issues following.
(i)We have demonstrated the ease with which chemical potentials such as Gibbs energy can be estimated in action space, using simple algorithms combining structural details of molecules and environmental conditions of temperature and pressure. Values for vibrational, rotational, and translational action allow corresponding entropies and entropic energies, especially Gibbs energy, to be calculated from the prevailing temperature. These easily computed estimates of maximum entropy at varying temperatures allow the feasibility of gas phase reactions forming hydrogen from water or ammonia to be examined. The algorithms provide a good starting point for research into sources of renewable energy but have value in general, including in education.(ii)The hypotheses advanced on catalysis and equilibrium are consistent with our previous research [4] on the Carnot cycle showing that the Gibbs energy (*G*, J) can be identified as an index of internal work comprising the field of energy referred to by Clausius as the ergal. We identify this field of statistically elevated mean values of quantum states physically sustaining the configuration of the material particles. This energy field sustaining its action complements the much smaller kinetic energy in an ideal system of unbound particles.(iii)Action mechanics also allows a useful revision of the Eyring transition state theory for the catalysis of chemical reactions. For uncatalysed reactions, it proposes a Boltzmann equilibrium between reactant molecules and their more or less activated species, with the reaction rate determined by the frequency of occurrence of the transition state, which is able to form products. This frequency depends uniquely on the strength of the bonding enthalpy and the temperature, determining the rate of reaction; the larger the barrier, the slower the rate. Uniquely, action mechanics proposes a reaction trajectory of quasi-equilibrium between substrates and species on binding sites on catalysts of similar Gibbs energy or chemical potential. We report the novel result that for simple reactions with unit steady-state activities of reactants and products, the ratio of the activated transition states must be equal to the equilibrium constant for the reaction. This result should be investigated with more complex reactions involving more than one reactant and product.(iv)Reactants bound on colloids are proposed to share their greater inertia and will therefore be subject to larger action impulses in collisions, either with free co-reactants or with co-reactants similarly bound on colloidal particles. The much lower density or frequency of catalytic colloids will ensure more sustained stresses in collisions, moving on straighter trajectories reflecting their greater inertial mass (*mr*), acting as microscopic anvils. Catalysis by enzymes remains mysterious; some enzymologists consider that many enzymes are “too big” for their binding functions alone [35], although the evolution of regulatory functions can also logically contribute to added size. However, if their inertial impetus in collisions is significant in forceful encounters with substrates, the size can be explained statistically in terms of enhancing the probability of increased stress for bound substrates in collisions. We suggest that the methods of action mechanics can also allow Gibbs energies to be calculated for such colloids, acting as forceful anvils, with respect to impulses affecting their vibrational, rotational and translational actions.(v)The ease of calculation of the number and frequency of virtual field quanta from translational and rotational action (Table 6), at least as mean values, was surprising. The mean value of the magnitude of virtual quanta shown in the action fields is effectively expressing the ergal. In principle, the values calculated indicate the field energy sustaining the action states of reactants and products in chemical reactions. It should be possible to obtain direct evidence of the frequency and the intensity of these quantum fields by using resonant detectors, since they lie in the range of infra-red, microwave, and radiowaves. There may be significant scope for controlling rates of reactions or even positions of equilibrium with sufficiently intense action fields such as laser beams.(vi)Recognising the magnitude of the field energy as the ergal may have profound importance for quantitative accounting of energy in many scenarios. This energy field is rarely considered or even hidden in areas as diverse as climate science, such as the potential or work energy stored in coherent cells of air such as anticyclones and cyclones, or in ‘calorie counting’ for human weight loss and nutrition. This fundamental principle of energy conservation was enunciated by Clausius almost two centuries ago in the equivalence of heat and work, but it is largely neglected. To what extent are the increasing obstructions to smooth air flow in modern cities and the ubiquitous wind farms of Europe and further afield causing local warming by reversible processes of internal work followed by turbulent release of radiant heat, particularly when masses of air activated in heat stored as vortical entropy [4] collide? The ability of the magnitude of changes in the ergal to release equivalent amounts of temperature-raising heat seems to have no role in the global circulation models on which the Intergovernmental Panel for Climate Change (IPCC) depends.

Considering the concept of Clausius’ ergal more generally, it is relevant to consider differences between solids, liquids, and gases. Lambert and Leff [36] have pointed out how in solids, standard entropy and enthalpy up to 298.15 K are very strongly correlated, with a slope of 0.0066 K^−1^, representing just the exponential terms in Equation (11) or (45), given that the mass of the solid is neither translating nor rotating and the scale of action shown in gases is absent.
Δ*H*^o = *T*^∫_o_*C*_p_(*T*) d*T*(53)

The increase of entropy by the absorption of heat at constant atmospheric pressure to reach temperature *T* reflects not only the thermal energy stored in the solid (*RT*ln[e^Σxf(x)^]) but also how that energy is stored, given the configurational work of even slight expansion against the atmosphere. For most vibrations in solids below 298.15 K, the vibrational entropy given by each degree of freedom *R*[*x*/(e*^x^* − 1) − ln(1 − e^−*x*^)]—effectively a work term—is negligible and contributes only slightly to the heat capacity. Heats of all phase changes including fusion and vaporisation contribute strongly to the entropy, including specific heats in liquids and gases (e.g., *RT*ln[e^7/2^]). General chemistry texts also usually emphasise “configurational entropy and the probability of locations”, which is a minor feature in solids [37]. Nevertheless, all solid chemicals or their solutions maintain a vapour pressure characteristic of the absolute temperature. For unsaturated solutions, the vapour pressure and corresponding fugacity (i.e., the ideal gas pressure) reflect the mole fraction of the substance. At equilibrium, the chemical potential, or the change in Gibbs energy with concentration of a chemical substance, is equal in all phases. This is also true of natural systems subject to variations in temperature as heat is absorbed or emitted in more or less steady-state conditions. 

It should be possible to use the fugacity of each chemical species as the effective pressure in each phase and to consider chemical reactivity on this basis, particularly at low pressures or concentrations where departures from the ideal gas law are minor. Even where they are larger, experimentally determined correction factors can be applied. The great advantage of fugacity is that it allows thermodynamic calculations yielding Δ*G^o^* as performed here, providing fugacity-based *K*_f_ values from the equality of −Δ*G^o^* and *RT*ln*K*_f_ to estimate the activity constant *K*γ from graphs of activity coefficients and then to calculate the actual partial pressures from *K*_p_ = *K*_f_/*K*_γ_.

In the case of biological molecules such as proteins and nucleic acids, it should also be possible to investigate these using the methods of action mechanics. For such large molecules, moments of inertia will be correspondingly greater, and there will be more internal motions such as bendings and rotations, all significantly contributing to the action and the corresponding ergal shown by decreasing chemical potential. For every identifiable chemical species and its kinetic energy, the larger associated quantum field or energy or ergal can be defined uniquely. The principle of maximum entropy is sometimes misunderstood as suggesting that entropy on Earth is always increasing as disorder. This is far from true. On the Earth’s surface and in its atmosphere, entropy actually stays approximately constant. The storage of heat by raising quantum states as occurs in the internal work of the ergal is a feedback mechanism that can absorb excessive sensible heat, raising the entropy that may re-emerge later as a reversible process. The increase in entropy is strongly constrained by radiation to space and limits on the diversity of heat sinks, which are all degrees of freedom.

The illustrations given here are all for gaseous reactions acting ideally. However, there is no reason that action mechanics should not be extended to the liquid or even the solid states. Fugacity theory [37] has shown that equilibrium between phases depends on the equality of vapour pressures of substances in equilibrium, whether in solid, liquid, or gaseous states. This suggests it should be possible to use fugacity as the effective pressure in each phase and to consider chemical reactivity on this basis. Most of all, a better understanding of action, entropy, and the supporting fields of energy should advance further innovative research. 

## Figures and Tables

**Figure 1 entropy-23-01056-f001:**
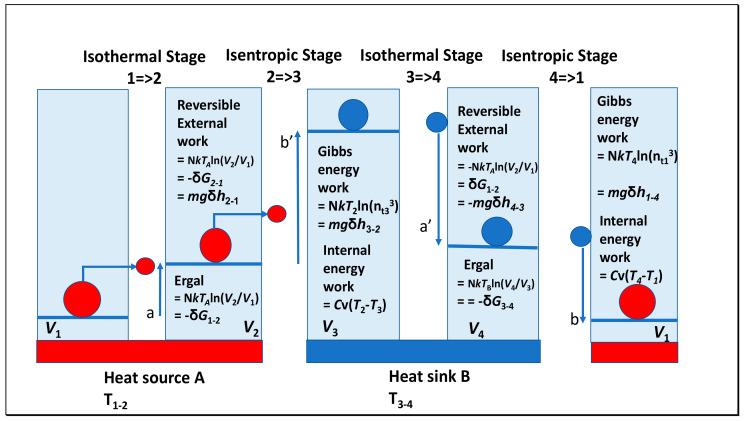
Variations in internal work (ergal) and external work (*δmgh*) in a Carnot cycle with argon as the working substance [4]. Note the equalization of internal and external work is a reversible process for an ideal gas. The isentropic processes are shown with translational quantum numbers nt3 and nt1 that do not vary in the adiabatic or isentropic stages when heat neither enters nor leaves. The Carnot cycle is neither a constant volume nor a constant pressure system therefore more heat than the kinetic energy is required for internal work or the ergal, that Carnot proposed as caloric (*a* − *a′*) or (*b′* − *b*) as the maximum work possible [4].

**Figure 3 entropy-23-01056-f003:**
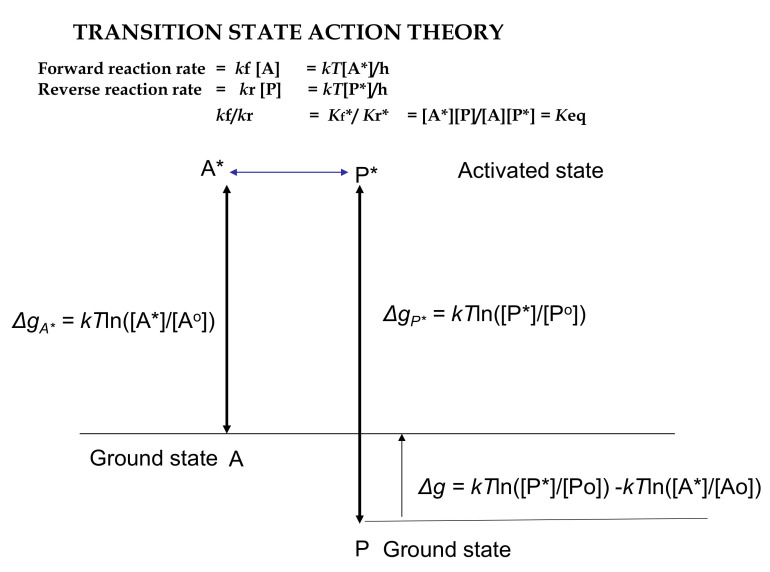
Action transition state diagram for conversion of *A* ↔ *P* showing excess work potential for conversion of *P** ⇒ *P* compared to *A** ⇒ *A*. This model was first proposed in 1992 [21]. Note that Δ*g* per molecule for activation is negative, given *A** << *A^o^*.

**Figure 4 entropy-23-01056-f004:**
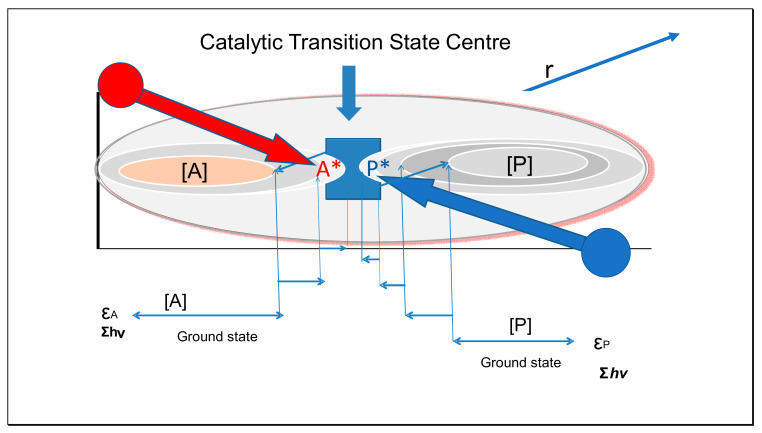
Radial action transition states on a Brownian catalyst, proposed to generate activation energy for reversible reactions *A* ↔ *P* by forceful (δ*mvr*) inertial collisions. Concentrations of *A** and *P** of similar chemical potential to bulk reactants *A* and *P* stabilise in potential wells, binding loosely. At constant temperature, the difference in ground-state energy ε_Po_ − ε_Ao_ represents the enthalpy and chemical potential changes for the reaction.

**Figure 5 entropy-23-01056-f005:**
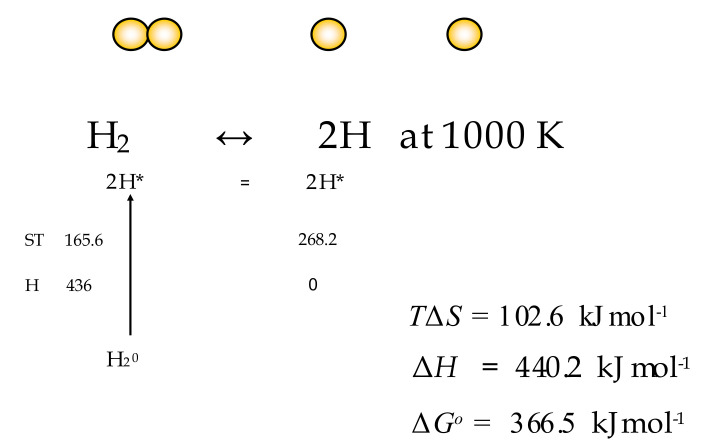
Gibbs energy change for the steady-state reaction dissociating unit activities of diatomic H_2_ to monatomic H. The change was estimated using the difference between cumulative bond energies and entropic energies. The bond energy at 298.15 K for HH is 436 kJ per mole. For ΔG^o^ of 366.5 kJ per mole, −lnK is 44.1 and K is 10^−19.1^. K is the factor indicating the ratio of the concentration of the forward transition species to the reverse species.

**Figure 6 entropy-23-01056-f006:**
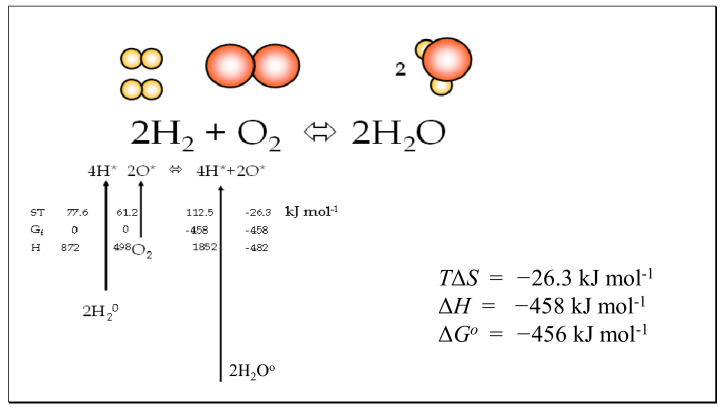
Standard Gibbs energy change for the steady-state reaction forming water from hydrogen and oxygen. Potential changes are usually estimated using standard tables or more simply from the differences between cumulative bond energies and entropic energies using action mechanics. Bond energies at 298.15 K for HH, OO, and H_2_O are 436, 498, and 926 kJ per mole, respectively. Given Δ*G^o^* is −456 kJ mol^−1^, then lnK is 183.8 and K = 10^79.8^, indicating the forward reaction is hugely faster than the reverse reaction.

**Figure 7 entropy-23-01056-f007:**
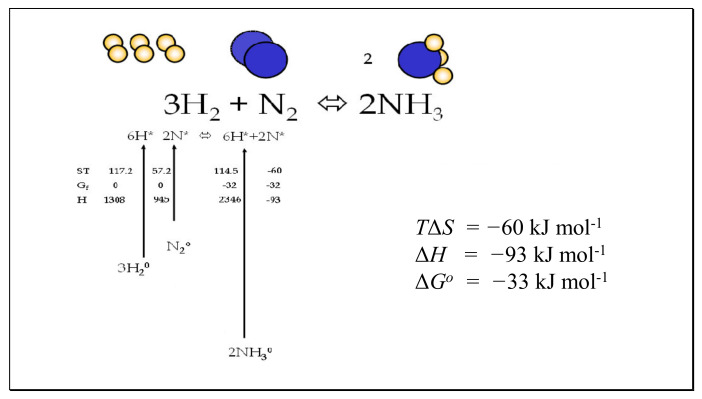
The Haber–Bosch fixation of dinitrogen for 1 atmosphere pressures of reactive species bond energies at 298.15 K for HH, NN, and NH_3_ are 436, 945, and 1173 kJ per mole, respectively. For Δ*G^o^* = −33 kJ mol^−1^ = −*RT*lnK, lnK = 13.3 and K = 10^5.78^. K is the factor indicating the ratio of the forward transition species to the reverse.

**Table 1 entropy-23-01056-t001:** Molecular properties used in calculations.

Gas	Mass Daltons	Bond Length pm	Principal Moments of Inertia *I*_a_, *I*_b_, *I*_c_ ×10^40^ g cm^2^	Vibration Frequencies cm^−1^	Degeneracy	Multiplicity Symmetry Q_e_, Rotational Symmetry σ
H	1.008	-	-		-	-
H_2_	2.016	74	-	4161	-	1, 2
N_2_	28.014	110	-	2358	-	1, 2
O_2_	16 + 16 = 32	121	-	1580	1	3, 2
NH_3_	14 + 1 + 1 + 1 = 17	101 101 101	2.9638 2.9638 4.5176	3337 950 3447 1627	1 1 1 1	1, 3
CO_2_	16 + 12 + 16 = 44	122 + 122 = 244	Linear	1388 667 2349	1 2 1	1, 2
H_2_O	16 + 1 + 1 = 18	74	1.024	3652	1	1, 2
		74	1.920	1595	1	
			2.947	3756	1	
N_2_O	14 + 14 + 16 = 44	NN 112 NO 119	Linear	2224 1285 589	1 1 2	1, 1
CH_4_	12 + 1 + 1 + 1 + 1 = 16	108 108 108 108	5.27 5.27 5.27	2914 1526 3020 1306	1 2 3 3	1, 12

Most of the data given is from Aylward and Findlay [17].

**Table 2 entropy-23-01056-t002:** Average bond enthalpies.

Bond	Δ*H* at 298 K kJ/mol	Δ*H* Ergs Per Molecule ×10^12^	Zero Point Vibrational Energy cm^−1^	Characteristic Temperature Rotation Θ_rot_ K	Characteristic Temperature Vibration Θ_vib_ K
H-H = H_2_	436	7.19234576	2079.307	85.4	6210
C-H = CH_4_	413	6.81293303			
N-H = NH_3_	391	6.45001650	7214.5		
O-H = H_2_O	463	7.01302636			
C=O = CO_2_	745	11.2844953		0.561	
C-O-	358	5.42259921			
C-N=	305	4.61981217			
O-O-	146	2.21145108			
O=O = O_2_	498	7.54316874	787.3797	2.07	2230
O-N=	201	3.04453196			
N=N-	418	6.33141472			
N_2_	945	14.3138443	1175.778	2.86	3374
N=O	607	9.19418358			

**Table 4 entropy-23-01056-t004:** Activated vibrational energy (ɛ) states for N_2_ and CO_2_ indicating translational action of molecules excited up to ɛ_vib_ = 5*hv*/2.

**640 K** **N_2_**	**N_0_/N_n_ = V_n_/V_o_**	**δɛ J**	***r*_n_/*r*_o_ = @_tn_/@_t0_**	**ɛ ex Action** ***kT*ln(n_tn_/n_to_)^3^**	**288 K** **N_2_**	**N_0_/N_n_ = V_n_/V_o_**	***r*_n_/*r*_o_ = @_tn_/@_t0_**	**ɛ ex Action** ***kT*ln(n_tn_/n_to_)^3^**
e^−δɛvib/kT^		×10^20^		×10^20^	e^−δɛvib^		×10^20^	×10^20^
e^−2*hv/kT*^	102,582	10.1954	46.812	10.1954	5*hv*/2	1.367e^11^	5151.40	10.1954
e^−*hv/kT*^	320.284	5.0977	6.842	5.0977	3*hv*/2	369605	77.771	5.0977
e^o^	1.000	0	1.000	0	*hv*/2	0	1.000	0
**640 K** **CO_2_**	**N_o_/N_n_ = V_n_/V_o_**	**δɛ ergs**	***r*_n_/*r*_o_ = @_tn_/@_t0_**	**ɛ ex Action** ***kT*ln(n_tn_/n_to_)^3^**	**288 K** **CO_2_**	**N_o_/N_n_ = V_n_/V_o_**	***r*_n_/*r*_o_ = @_tn_/@_t0_**	**ɛ ex Action** ***kT*ln(n_tn_/n_to_)^3^**
δɛ_vib_					δɛ_vib_		×10^13^	×10^13^
e^−2*hv/kT*^	20.065	2.6499	2.717	2.6499	e^−2*hv/kT*^	783.99	9.221	2.6499
e^−*hv/kT*^	4.479	1.3250	1.648	1.3250	e^−*hv/kT*^	28.000	3.037	1.3250
e^o^	1.000	0	1.000	0	e^o^	1.000	1.000	0

**Table 5 entropy-23-01056-t005:** Temperature variation of transition state Gibbs energies for dissociation of hydrogen molecules to hydrogen atoms.

Temp K	Entropy (^o^*S*) at 1 atm J/K	*ST* kJ/mole	Entropy (*S*) at 1 atm J/K	*ST* kJ/mole	−*T*Δ*S*	−Δ*H* H_2_=>2H = 436 kJ	Gibbs Change Δ*G*
	H	2H	H_2_	H_2_			
1000	134.088	268.176	165.562	165.562	102.614	440.157	−366.543
2000	148.496	593.984	185.733	371.466	222.518	444.314	−216.796
3000	156.924	941.544	197.533	592.599	348.445	448.471	−99.526
4000	162.904	1303.232	205.904	823.616	479.616	481.628	−2.012
5000	167.542	1675.420	212.348	1061.990	613.010	456.785	+98.225
6000	171.332	2055.984	217.703	1306.221	749.763	460.942	+230.821
7000	174.536	2443.504	222.189	1555.323	888.181	465.099	+365.082
8000	177.312	2836.992	226.075	1808.600	1028.342	469.256	+501.086
9000	179.760	3235.680	229.503	2065.520	1170.160	473.413	+648.747

**Table 6 entropy-23-01056-t006:** Partitioning of internal Gibbs energy (J ergal) into translational and rotational virtual quanta.

Temp	Translational Ergal H J	Translational Virtual Quanta	Mean Energy H *hω*	Translational Ergal H_2_	Translational Virtual Quanta	Mean Energy H_2_ *hω*	Rotational Ergal	Rotational Virtual Quanta	Mean Energy H_2_ *hω*
K	*g*_t_ ×10^19^	n_t_	×10^27^	*g*_t_ ×10^19^	n_t_	×10^21^	*g*_r_ ×10^20^	j_r_	×10^20^
1000	1.8815	203.8	22.23	2.0250	288.3	0.702	4.8324	3.385	1.428
2000	4.2414	363.2	8.853	4.5285	513.7	0.882	9.4958	4.788	1.983
3000	6.7820	509.3	5.134	7.2127	720.2	1.001	11.7813	5.864	2.009
4000	9.4399	647.2	1.459	10.0141	915.3	1.094	17.2972	6.771	2.555
5000	12.1849	779.5	1.563	12.9028	1102.4	1.170	23.1626	7.570	3.060
6000	14.9996	907.4	1.653	15.8608	1283.3	1.236	29.3055	8.292	3.534
7000	17.8720	1031.8	1.732	18.8769	1459.2	1.293	35.6788	8.957	3.983
8000	20.7938	1153.2	1.803	21.9422	1630.9	1.345	42.2504	9.575	4.413
9000	23.7589	1272.2	1.868	25.0508	1799.1	1.392	48.9963	10.156	4.824

**Table 7 entropy-23-01056-t007:** Temperature variation for transition state Gibbs energies for dissociation of water into oxygen and hydrogen.

K Temp	H_2_O, *S*, *ST*	O_2_, *S*, *ST*	H_2_, *S*, *ST*	*T*Δ*S* 2H_2_ + O_2_	−Δ*H* <> 2H_2_O	Δ*G=* Δ*H* − *T*Δ*S*
	J/K, kJ/mole	J/K, kJ/mole	J/K, kJ/mole	kJ/mole	444 @ 0	
1000	232.397, 232.397	243.403, 243.403	165.671, 165.671	109.951	463.112	−353.161
2000	264.043, 528.086	267.976, 521.005	187.249, 374.498	213.829	474.273	−260.444
3000	285.241, 855.724	283.116, 849.348	201.024, 603.071	344.042	485.208	−141.166
4000	309.471, 1237.885	293.795, 1175.178	211.215, 844.858	389.124	496.715	−107.591
5000	313.582, 1567.910	302.104, 1510.519	219.280, 1096.402	567.503	508.680	+58.823
6000	323.942, 1943.651	308.904, 1853.422	225.943, 1335.658	637.436	520.770	+116.666
7000	332.760, 2329.319	314.658, 2202.607	231.613, 1621.291	786.551	532.904	+253.647
8000	340.431, 2723.448	319.646, 2557.165	236.545, 1892.359	894.987	545.178	+349.809
9000	347.217, 3124.451	324.047, 2914.642	240.907, 2168.167	1002.074	557.541	+444.533
10,000	353.299, 3533.00	327.985, 3279.850	244.818, 2448.176	1110.218	569.883	+540.335

**Table 8 entropy-23-01056-t008:** Change in enthalpies in water formation.

Temperature	H_2_O kJ/mol	O_2_	H_2_	Δ*H*
1000	37.141	34.598	29.398	−19.112
2000	83.468	73.149	62.030	−30.273
3000	130.953	111.084	96.015	−41.208
4000	177.954	148.719	129.952	−52.715
5000	224.562	186.366	163.719	−64.680
6000	270.918	223.898	197.354	−76.770
7000	317.144	261.396	230.898	−88.904
8000	363.201	298.824	264.378	−101.178
9000	409.213	336.339	297.814	−113.541
10,000	455.171	373.793	331.216	−125.883

**Table 9 entropy-23-01056-t009:** Variation with temperature of Gibbs energy for Haber–Bosch reaction.

Temp. K	NH_3_, *S*, *ST*	N_2_, *S*, *ST*	H_2_, *S*, *ST*	*T*Δ*S*	−Δ*H* 93 at 0 K	−Δ*G*
	J/K, kJ/mole	J/K, kJ/mole	J/K, kJ/mole	kJ/mole	kJ/mole	kJ/mole
400	203.727, 81.491	200.189, 80.076	138.898, 55.559	83.771	97.924	14.153
500	212.331, 106.166	206.683, 103.342	145.391, 72.696	109.098	96.996	−12.102
600	219.750, 131.850	211.989, 127.193	150.697, 90.418	134.747	95.300	−39.447
700	226.328, 158.430	216.475, 151.533	155.183, 108.628	160.557	93.000	−67.557
800	232.277, 185.822	220.361, 176.289	159.069, 127.255	186.410	90.094	−93.316
900	237.732, 213.959	223.788, 201.409	162.496, 146.246	212.229	86.630	−125.599
1000	242.789, 242.789	226.854, 226.854	165.562, 165.562	237.962	82.683	−155.279

**Table 10 entropy-23-01056-t010:** Kinetic enthalpy for the Haber process.

	NH_3_	N_2_	H_2_	Δ*H*
400	14.998	11.640	11.640	−4.924
500	19.842	14.550	14.550	−3.996
600	25.040	17.460	17.460	−2.300
700	30.555	20.370	20.370	0
800	36.373	23.280	23.280	2.906
900	42.470	26.190	26.190	6.370
1000	48.810	29.101	29.101	10.317

**Table 11 entropy-23-01056-t011:** Entropy and entropic energy changes, including vibrational heat capacity.

Temp K	NH_3_, *S*, *ST*	N_2_, *S*, *ST*	H_2_, *S*, *ST*	*T*Δ*S* + Δ*C*v
	J/K, kJ/mole	J/K, kJ/mole	J/K, kJ/mole	kJ/mole
400 *	207.964, 83.186	200.189, 80.076	138.898, 55.559	−80.381
500	218.758, 109.379	206.683, 103.342	145.391, 72.696	−102.672
600	228.224, 136.934	211.989, 127.193	150.697, 90.418	−124.979
700	236.721, 165.705	216.475, 151.533	155.183, 108.628	−146.007
800	244.485, 195.588	220.361, 176.289	159.069, 127.255	−166.878
900	251.663, 226.497	223.788, 201.409	162.496, 146.246	−196.153
1000	258.342, 258.342	226.854, 226.854	165.562, 165.562	−206.856

* Ammonia triple point is near 400 K, condensing at 237 K at 1 atm, presenting an interesting problem.

## Data Availability

All data used is contained within the article.

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
