# Peer review of "Partitioning Entropy with Action Mechanics: Predicting Chemical Reaction Rates and Gaseous Equilibria of Reactions of Hydrogen from Molecular Properties"

_entropy, 2021, doi:10.3390/e23081056_

Round 1
Reviewer 1 Report
In this article, the authors after an interesting introduction citing several historical papers, develop the theory of what they call “transition state action version of Eyring’s absolute transition state theory for chemical reactions”. Their methods named “action mechanics” are briefly discussed and then applied to three illustrative case studies, i) dissociation of the hydrogen molecule, ii) association/dissociation of water, and iii) ammonia production from hydrogen and nitrogen molecules.
I understand that there is a previous publication in which the investigators introduce their methods of action mechanics, nevertheless, I believe that the authors should make the effort to explain what are the benefits of employing actions in order to understand the thermodynamics and kinetics of chemical reactions. Specifically,
1) The assignment of actions to basic molecular motions, translational-rotational-vibrational, and their calculations for the case studies are based on standard statistical mechanical formulae, extracted from exact solutions of solvable models in quantum and classical mechanics. These are the multi-dimensional harmonic oscillator, free rigid rotors (symmetric and asymmetric), and the translational motion of the center of mass.
To my opinion, a future reader of the article will need convincing arguments than a sentence like
“Compared to the reductionist factors of temperature, volume or pressure normally applied in statistical mechanics, molecular action is holistic, displaying the physical form of universal reality appealing to imagination.”
I understand that there is a plan to extend the action theory for studying complex systems encountered in environmental and biological chemistry, such as enzymatic reactions. Thus, explaining what new action mechanics bring to thermodynamics is imperative.
2) Transition state theory for chemical reactions is a dynamical theory and it has never been stopped to attract the interest of the researchers since its first invention (for example, Pechukas, P. and McLafferty, F. J., On transition state theory and the classical mechanics of collinear collisions, The Journal of Chemical Physics, 1973, vol. 58, pp. 1622-1625.).
As a matter of fact, it was only at the end of the twentieth century that we learned how to mathematically define the transition state as time-invariant submanifolds in molecular phase space, and a precise definition of activated complex was given for polyatomic molecules.
However, one has to admit that the numerical calculation of these dynamical objects is not an easy task. Molecular potential energy hypersurfaces are complex nonlinear functions and transition states more often are identified as saddles on these hypersurfaces in the coordinate configuration space. Quantum chemistry methods have shown that in most cases, the geometry of the saddle differs from the geometry of reactant molecules more than the extension of a few bonds.
The authors of the article in their effort to elucidate some misunderstood points in transition state theory introduce the existence of an equilibrium between forward and reverse reaction activated complexes
“Realistically, a standard unit concentration transition state cannot be in equilibrium or even in quasi-equilibrium with its ground state at the same unit concentration. In action mechanics, the effective – ΔG factor in the reaction must be the Gibbs energy difference for the transition *A_f  *P_r from forward and reverse directions and not that for the transition *A_o  A.”
Elementary chemical reactions, such as the photodissociation of a molecule, involve a single transition state that separates reactants from products. Thus, composite chemical reactions are defined as aggregates of elementary reactions with more than one transition state. So, what do the investigators mean when they introduce equilibrium between *A and *P? Is it a composite system with a metastable structure between them?
Also, in an elementary chemical reaction with a single transition state *A and *P should be the same species and then one has to take the equilibrium of *A_o  A. The authors must discuss these issues.
3) Finally, a more technical remark that concerns the notation used by the authors. They understand that among the readership chemists are included. In the chemical literature, reversible chemical reactions are usually denoted by two arrows with opposite directions, but the authors prefer symbols like those in equations (38) and (52). Any particular reason?
I find the symbol of action also strange. Usually, angular momentum is denoted by l or L and more generally actions by the letter J or I.
The authors should examine the above remarks before considering the article for publication.
Author Response
Dear Reviewer 1,
Thank you for your comprehensive review and your precise suggestions which we have used to improve the manuscript.
- We accept that your criticism regarding a need for better description of the methods and how that action mechanics has merit. The aims and purposes of the article are now given in the first line of the Abstract and the first paragraph of the Introduction. We have also rewritten almost completely the Conclusions, listing all our significant findings. With respect to reactions of hydrogen, we have explained that the non-equilibrium ΔG values at different temperatures given in the tables 4-6 are steady-state results indicating potential for work, showing the temperatures where work can be performed. Steady states with significant concentrations of reactants and products like chemostats are more typical of real-world processes where equilibrium conditions are usually not desirable, lacking work potential. We have partly revised the Methods, adding a glossary to Table 2 to clarify terminology and also modified text, shown in track changes.
- Regarding transition state theory, we accept there is a formidable challenge that you present in defining potential energy surfaces. However, that is not our intention and we prefer to keep the discussion as simple as possible; our main purpose is to clarify understanding of the transition processes using the action mechanics approach to transitions in Gibbs energy. This gives different results to those published, reversing the process as entropy is increased in the activated state and Gibbs energy reduced. We introduce a new action mechanics proposal that radial action impulses in collisions on colloids with higher inertial mass (mr) have a significant role in achieving activated transition states, decreasing internal Gibbs energy by increasing the amplitude of vibrations. This mechanism can explain the “bigness” of enzymes, hitherto unexplained. We leave an analysis of energy surfaces for future work. Thank you for mentioning the relevant paper in J.Chem.Phys., that we have included in our discussion suggesting a way forward.
- Thanks for the question regarding detail of activated states and reversible reactions. For equilibrium at equal concentrations of activated species A* and P*, we do not propose that the reactant and the product involve composite metastable transition states, either free or on catalysts. We have added text to explain that these species should occur independently. We agree with Eyring’s original suggestion that the probability of transition is large once the activation energy has been achieved, now stated in the text. Forward and reverse reactions should occur independently as a function of reactant and product activities. Consideration of both directions would only be needed near equilibrium. Another of our new findings, at least for the single reactant-product reaction, is that the respective activated transition activities should have the same ratio as the equilibrium constant if the reactant and the product are both maintained in a steady state at standard unit activity of 1 molal or atm pressure. This finding is applied as the basis for the ΔG values calculated for the hydrogen reactions.
- We accept your criticism regarding unconventional notation has merit. Separate arrows for forward and reverse reactions would be more consistent with our argument made in response (3) of their independence, so we have simplified the notation. Regarding your hesitation regarding our use of @ as the relative action symbol -- apart from its suggestive appearance we prefer a special symbol to distinguish this scalar version of action from the vector angular momentum. Obviously, its meaning on the Underwood typewriter as “at a rate of” is also attractive. For philosophers, the @ symbol is taken as one for actual suiting the physically realistic role we propose. It also serves to indicate the quantum state nature of translational, rotational and vibrational actions, leading to a complementary role for kinetics such as in harmonic oscillators and their complementary action analogue as field energy. For more clarity in this important area, we have added units with physical dimensions in square brackets whenever different properties are first mentioned.
- In making our revision we also realized that there is an opportunity to discuss the principle of maximum entropy in this article. Clausius’ ergal helps to explain how to establish a maximum entropy value and how it is constrained by quantum processes that action mechanics relies on.
- We have replaced the text you selected as unconvincing, replacing it with a more modest statement as follows:
- (i) Compared to the reductionist factors of temperature, volume or pressure normally applied in statistical mechanics, molecular action is holistic, displaying the physical form of universal reality appealing to imagination.”
Compared to the reductionist factors of temperature, volume or pressure normally applied in equations for statistical mechanics, molecular action is holistic, combining these terms in a physical property we claim to be realistic and a useful function for developing theory.
Unlike angular momentum, which indicates a continuous intensity of activity, action is more akin to an operation and the @ symbol becomes an operator since it involves relative motion. We expect this feature to emerge over time, so we prefer to have a separate symbol, easily accessible.
(ii) Realistically, a standard unit concentration transition state cannot be in equilibrium or even in quasi-equilibrium with its ground state at the same unit concentration. In action mechanics, the effective –ΔG factor in the reaction must be the Gibbs energy difference for the transition *Af ↔ *Pr from forward and reverse directions and not that for the transition *Ao ↔ A.
We stand by this statement, its significance already being referred to above in point (3). A reactant and a product at unit concentration are extremely likely to be at equilibrium concentrations unless they have exactly the same energy. Our analysis has shown in Section 3.4 that forward and reverse transition state concentrations are equal at equilibrium, explaining the lack of net reaction. Under non-equilibrium conditions, we have shown (uniquely, we believe) that, when reactants and products are in standard states, that the ratio [A*]/[P*] must be equal the equilibrium K value, though not at other concentrations. This is the basis of the calculations for the reactions of hydrogen, all pressures being 1 atm, shown in Tables 4-6.
Thank you for your comments. They have enabled us to raise the potential impact of the article.
Reviewer 2 Report
I think the article "Partitioning Entropy with Action Mechanics: Predicting Chemical Reaction Rates and Gaseous Equilibria of Reactions of Hydrogen from Molecular Properties".
Authors: Ivan Robert Kennedy and Migdat Hodzic
The article is in the context of entropy journal. I think the article must improve the presentation of their results, equations and so on. Not all equations are familiar to the reader, so a better explanation would increase readers understanding. Other point, author forgot the nomenclature, with this much symbols .
I have a discussion, is Helmholtz and Gibbs fuction are potential to perform work, or their variation represents these phisical properties?
Authors must improve the tables organization, figures and its quality.
Therefore, I recomend major reviews for them bein possible to carry out a proper analysis of the article.
Author Response
Dear Reviewer 2,
Thank you for your helpful critique of this manuscript and your opening comment that it includes topics suitable for publication in Entropy, subject to major revision. We accept that major improvements were needed in the ms.
Our responses to your suggestions for improvement on all counts follow:
1) We have revisited all the equations with a view to providing better explanations, shown in track changes in the revised manuscript. For nomenclature, to aid understanding of all action and energy terms, we have included physical units such as Joule.sec (J.sec) or Joules (J) in square brackets. We have also added a brief glossary of the terminology in Table 1, to assist the reader.
2) We have revised all the figures and tables as requested, with regard to clarity and quality. Figures 1, 3, 5, 6, 7 have all been redrawn, removing all the untidy text except energy values, now simplified in the legend.
3) Absolute Gibbs (G) and Helmholtz (A) energies as calculated in the manuscript have the usual classical meanings, with values indicating the potential to perform work, with suitable inputs of thermal energy. Where appropriate, we have used ΔG values when considering differences in state in chemical reactions as in Tables 4-6. We have explained that these data represent steady-state reactions, giving Gibbs energy changes per mole holding reactants and products at constant activity, like many reactions in real systems. It is unusual to calculate absolute values of A or G per molecule or per mole, but the definition of relative action in action mechanics makes this possible. We have emphasized that a unique advantage of the methods described is they show how A and G can be calculated directly from molecular properties, under stated environmental properties of pressure and temperature. We have included new explanations how their variation occurs from changes in these physical conditions as well as in chemical reactions when bonding energies are altered as new bonds and conformations are formed. Thank you for this point of discussion.
4) From your comments we have made numerous changes to the text, including expressing aims more clearly in the Introduction and rewriting the Conclusion, listing the significance of the main findings in the article.
Thank you for your assistance.
Round 2
Reviewer 1 Report
Certainly, the revised manuscript has significantly been improved. The authors have provided clarifications with respect to the raised points, and to some extent, the notation has been improved.
Action mechanics bring a different point of view in computing molecular thermodynamic quantities, but still, it remains to be seen if they work in non-ideal systems.
Reviewer 2 Report
This is the second round of review of the article: "Partitioning Entropy with Action Mechanics: Predicting Chemical Reaction Rates and Gaseous Equilibria of Reactions of Hydrogen from Molecular Properties"
The significance of the article is self explained, and, again, it is very interesting to read articles with fundamental of thermodynamics. Therefore, with the authors improvement I think the article is suitable for publication.
Regarding the Gibbs energy, now it is clear in the text that G = H - TS is a thermodynamic property and when we calculated its variation over two different states it has this physical meaning of maximum available work. I am only writing this to see if authors agree.
For the remain of the manuscript I think it is suitable for publication and entropy and may call much atention from readers that are looking for these thermodynamic properties in order to study the hydrogen as a fuel. And from basic thermodynamics point of view it is a complete aplication in order to obtain all these properties.